# A-NeSI: A Scalable Approximate Method for Probabilistic Neurosymbolic Inference

**Emile van Krieken**
University of Edinburgh
Vrije Universiteit Amsterdam
Emile.van.Krieken@ed.ac.uk

**Thiviyan Thanapalasingam**
University of Amsterdam

**Jakub M. Tomczak**
Eindhoven University of Technology

**Frank van Harmelen, Annette ten Teije**
Vrije Universiteit Amsterdam

## Abstract

We study the problem of combining neural networks with symbolic reasoning. Recently introduced frameworks for Probabilistic Neurosymbolic Learning (PNL), such as DeepProbLog, perform exponential-time exact inference, limiting the scalability of PNL solutions. We introduce *Approximate Neurosymbolic Inference* (A-NeSI): a new framework for PNL that uses neural networks for scalable approximate inference. A-NeSI 1) performs approximate inference in polynomial time without changing the semantics of probabilistic logics; 2) is trained using data generated by the background knowledge; 3) can generate symbolic explanations of predictions; and 4) can guarantee the satisfaction of logical constraints at test time, which is vital in safety-critical applications. Our experiments show that A-NeSI is the first end-to-end method to solve three neurosymbolic tasks with exponential combinatorial scaling. Finally, our experiments show that A-NeSI achieves explainability and safety without a penalty in performance.

## 1 Introduction

Recent work in neurosymbolic learning combines neural perception with symbolic reasoning [52, 38], using symbolic knowledge to constrain the neural network [19], to learn perception from weak supervision signals [36], and to improve data efficiency [5, 57]. Many neurosymbolic methods use a differentiable logic such as fuzzy logics [5, 17] or probabilistic logics [35, 57, 16]. We call the latter *Probabilistic Neurosymbolic Learning (PNL)* methods. PNL methods add probabilities over discrete truth values to maintain all logical equivalences from classical logic, unlike fuzzy logics [54]. However, performing inference requires solving the *weighted model counting (WMC)* problem, which is computationally exponential [13], significantly limiting the kind of tasks that PNL can solve.

We study how to scale PNL to exponentially complex tasks using deep generative modelling [51]. Our method called *Approximate Neurosymbolic Inference* (A-NeSI), introduces two neural networks that perform approximate inference over the WMC problem. The *prediction model* predicts the output of the system, while the *explanation model* computes which worlds (i.e. which truth assignment to a set of logical symbols) best explain a prediction. We use a novel training algorithm to fit both models with data generated using background knowledge: A-NeSI samples symbol probabilities from a prior and uses the symbolic background knowledge to compute likely outputs given these probabilities. We train both models on these samples. See Figure 1 for an overview.

A-NeSI combines all benefits of neurosymbolic learning with scalability. Our experiments on the Multi-digit MNISTAdd problem [35] show that, unlike other approaches, A-NeSI scales almost

| Forward process | Sample belief | Choose example | | Reason |
| --- | --- | --- | --- | --- |

$p(\mathbf{P}, \mathbf{w}, \mathbf{y})$ $\qquad$ $p(\mathbf{P})\cdots\!\blacktriangleright\mathbf{P}\longrightarrow p(\mathbf{w}|\mathbf{P})\cdots\!\blacktriangleright\mathbf{w}\longrightarrow c\longrightarrow\mathbf{y}$

$q_{\boldsymbol{\phi}}(\mathbf{w}, \mathbf{y}|\mathbf{x})$ $\quad \mathbf{x}\longrightarrow f_{\boldsymbol{\theta}}\longrightarrow\mathbf{P}\longrightarrow q_{\boldsymbol{\phi}_p}(\mathbf{y}|\mathbf{P})\cdots\!\blacktriangleright\mathbf{y}\longrightarrow q_{\boldsymbol{\phi}_e}(\mathbf{w}|\mathbf{y}, \mathbf{P})\cdots\!\blacktriangleright\mathbf{w}$

| Inference model | Perceive | Predict | Explain (optional) |
| --- | --- | --- | --- |

Figure 1: Overview of A-NESI. The forward process samples a belief $\mathbf{P}$ from a prior, then chooses a world $\mathbf{w}$ for that belief. The symbolic function $c$ computes its output $\mathbf{y}$. The inference model uses the perception model $f_{\boldsymbol{\theta}}$ to find a belief $\mathbf{P}$, then uses the prediction model $q_{\boldsymbol{\phi}}$ to find the most likely output $\mathbf{y}$ for that belief. If we also use the optional explanation model, then $q_{\psi}$ explains the output.

linearly in the number of digits and solves MNISTAdd problems with up to 15 digits while maintaining the predictive performance of exact inference. Furthermore, it can produce explanations of predictions and guarantee the satisfaction of logical constraints using a novel symbolic pruner.

The paper is organized as follows. In Section 3, we introduce A-NESI. Section 3.1 presents scalable neural networks for approximate inference. Section 3.2 outlines a novel training algorithm using data generated by the background knowledge. Section 3.2.4 extends A-NESI to include an explanation model. Section 3.3 extends A-NESI to guarantee the satisfaction of logical formulas. In Section 4, we perform experiments on three Neurosymbolic tasks that require perception and reasoning. Our experiments on Multi-digit MNISTAdd show that A-NESI learns to predict sums of two numbers with 15 handwritten digits, up from 4 in competing systems. Similarly, A-NESI can classify Sudoku puzzles in $9 \times 9$ grids, up from 4, and find the shortest path in $30 \times 30$ grids, up from 12.

## 2 Problem setup

First, we introduce our inference problem. We will use the MNISTAdd task from [35] as a running example. In this problem, we must learn to recognize the sum of two MNIST digits using only the sum as a training label. Importantly, we do not provide the labels of the individual MNIST digits.

### 2.1 Problem components

We introduce four sets representing the spaces of the variables of interest.

1. $X$ is an input space. In MNISTAdd, this is the pair of MNIST digits $\mathbf{x} = (\boxed{5}, \boxed{8})$.
2. $W$ is a structured space of $k_W$ different discrete variables $w_i$, each with their own domain. Its elements $\mathbf{w} \in W = W_1 \times ... \times W_{k_W}$ are *worlds* or *concepts* [59] of some $\mathbf{x} \in X$. For $(\boxed{5}, \boxed{8})$, the correct world is $\mathbf{w} = (5, 8) \in \{0, ..., 9\}^2$.
3. $Y$ is a structured space of $k_Y$ discrete variables $y_i$. Elements $\mathbf{y} \in Y = Y_1 \times ... \times Y_{k_Y}$ represent the output of the neurosymbolic system. Given world $\mathbf{w} = (5, 8)$, the sum is 13. We decompose the sum into individual digits, so $\mathbf{y} = (1, 3) \in \{0, 1\} \times \{0, ..., 9\}$.
4. $\mathbf{P} \in \Delta^{|W_1|} \times ... \times \Delta^{|W_{k_W}|}$, where each $\Delta^{|W_i|}$ is the probability simplex over the options of the variable $w_i$, is a *belief*. $\mathbf{P}$ assigns probabilities to different worlds with $p(\mathbf{w}|\mathbf{P}) = \prod_{i=1}^{k_W} \mathbf{P}_{i, w_i}$. That is, $\mathbf{P}$ is a parameter for an independent categorical distribution over the $k_W$ choices.

We assume access to some *symbolic reasoning function* $c : W \to Y$ that deterministically computes the (structured) output $\mathbf{y}$ for any world $\mathbf{w}$. This function captures our background knowledge of the problem and we do not impose any constraints on its form. For MNISTAdd, $c$ takes the two digits $(5, 8)$, sums them, and decomposes the sum by digit to form $(1, 3)$.

### 2.2 Weighted Model Counting

Together, these components form the *Weighted Model Counting (WMC)* problem [13]:

$$p(\mathbf{y}|\mathbf{P}) = \mathbb{E}_{p(\mathbf{w}|\mathbf{P})}[\mathbb{1}_{c(\mathbf{w})=\mathbf{y}}] \tag{1}$$

The WMC counts the *possible* worlds $\mathbf{w}$[1] for which $c$ produces the output $\mathbf{y}$, and weights each possible world $\mathbf{w}$ with $p(\mathbf{w}|\mathbf{P})$. In PNL, we want to train a perception model $f_\theta$ to compute a belief $\mathbf{P} = f_\theta(x)$ for an input $\mathbf{x} \in X$ in which the correct world is likely. Note that *possible* worlds are not necessarily *correct* worlds: $\mathbf{w} = (4, 9)$ also sums to 13, but is not a symbolic representation of $\mathbf{x} = (\text{\fbox{5}}, \text{\fbox{8}})$.

Given this setup, we are interested in efficiently computing the following quantities:

1. $p(\mathbf{y}|\mathbf{P} = f_\theta(\mathbf{x}))$: We want to find the most likely outputs for the belief $\mathbf{P}$ that the perception network computes on the input $\mathbf{x}$.

2. $\nabla_{\mathbf{P}} p(\mathbf{y}|\mathbf{P} = f_\theta(\mathbf{x}))$: We want to train our neural network $f_\theta$, which requires computing the gradient of the WMC problem.

3. $p(\mathbf{w}|\mathbf{y}, \mathbf{P} = f_\theta(\mathbf{x}))$: We want to find likely worlds given a predicted output and a belief about the perceived digits. The probabilistic logic programming literature calls $\mathbf{w}^* = \arg\max_{\mathbf{w}} p(\mathbf{w}|\mathbf{y}, \mathbf{P})$ the *most probable explanation (MPE)* [50].

## 2.3   The problem with exact inference for WMC

The three quantities introduced in Section 2.2 require calculating or estimating the WMC problem of Equation 1. However, the exact computation of the WMC is #P-hard, which is above NP-hard. Thus, we would need to do a potentially exponential-time computation for each training iteration and test query. Existing PNL methods use probabilistic circuits (PCs) to speed up this computation [60, 28, 35, 57]. PCs compile a logical formula into a circuit for which many inference queries are linear in the size of the circuit. PCs are a good choice when exact inference is required but do not overcome the inherent exponential complexity of the problem: the compilation step is potentially exponential in time and memory, and there are no guarantees the size of the circuit is not exponential.

The Multi-digit MNISTAdd task is excellent for studying the scaling properties of WMC. We can increase the complexity of MNISTAdd exponentially by considering not only the sum of two single digits (called $N = 1$) but the sum of two numbers with multiple digits. An example of $N = 2$ would be $\text{\fbox{52}} + \text{\fbox{83}} = 135$. There are 64 ways to sum to 135 using two numbers with two digits. Contrast this to summing two digits: there are only 6 ways to sum to 13. Each increase of $N$ leads to 10 times more options to consider for exact inference. Our experiments in Section 4 will show that approximate inference can solve this problem up to $N = 15$. Solving this using exact inference would require enumerating around $10^{15}$ options for each query.

# 3   A-NeSI: Approximate Neurosymbolic Inference

Our goal is to reduce the inference complexity of PNL to allow training neurosymbolic models on much more complex problems. To this end, in the following subsections, we introduce *Approximate Neurosymbolic Inference* (A-NeSI). A-NeSI approximates the three quantities of interest from Section 2, namely $p(\mathbf{y}|\mathbf{P})$, $\nabla_{\mathbf{P}} p(\mathbf{y}|\mathbf{P})$ and $p(\mathbf{w}|\mathbf{y}, \mathbf{P})$, using neural networks. We give an overview of our method in Figure 1.

## 3.1   Inference models

A-NeSI uses an *inference model* $q_{\phi, \psi}$ defined as

$$q_{\phi, \psi}(\mathbf{w}, \mathbf{y}|\mathbf{P}) = q_\phi(\mathbf{y}|\mathbf{P}) q_\psi(\mathbf{w}|\mathbf{y}, \mathbf{P}). \tag{2}$$

We call $q_\phi(\mathbf{y}|\mathbf{P})$ the *prediction model* and $q_\psi(\mathbf{w}|\mathbf{y}, \mathbf{P})$ the *explanation model*. The prediction model should approximate the WMC problem of Equation 1, while the explanation model should predict likely worlds $\mathbf{w}$ given outputs $\mathbf{y}$ and beliefs $\mathbf{P}$. One way to model $q_{\phi, \psi}$ is by considering $W$ and $Y$ as two sequences and defining an autoregressive generative model over these sequences:

$$q_\phi(\mathbf{y}|\mathbf{P}) = \prod_{i=1}^{k_Y} q_\phi(y_i|\mathbf{y}_{1:i-1}, \mathbf{P}), \quad q_\psi(\mathbf{w}|\mathbf{y}, \mathbf{P}) = \prod_{i=1}^{k_W} q_\psi(w_i|\mathbf{y}, \mathbf{w}_{1:i-1}, \mathbf{P}) \tag{3}$$

---

[1]Possible worlds are also called 'models'. We refrain from using this term to prevent confusion with '(neural network) models'.

**Algorithm 1** Compute inference model loss

    fit prior $p(\mathbf{P})$ on $\mathbf{P}_1, ..., \mathbf{P}_k$
    $\mathbf{P} \sim p(\mathbf{P})$
    $\mathbf{w} \sim p(\mathbf{w}|\mathbf{P})$
    $\mathbf{y} \leftarrow c(\mathbf{w})$
    **return** $\phi + \alpha \nabla_\phi \log q_\phi(\mathbf{y}|\mathbf{P})$

**Algorithm 2** A-NESI training loop

    **Input:** dataset $\mathcal{D}$, params $\boldsymbol{\theta}$, params $\phi$
    `beliefs`$\leftarrow []$
    **while** not converged **do**
        $(\mathbf{x}, \mathbf{y}) \sim \mathcal{D}$
        $\mathbf{P} \leftarrow f_{\boldsymbol{\theta}}(\mathbf{x})$
        update `beliefs` with $\mathbf{P}$
        $\phi \leftarrow$ **Algorithm 1**(`beliefs`, $\phi$)
        $\boldsymbol{\theta} \leftarrow \boldsymbol{\theta} + \alpha \nabla_{\boldsymbol{\theta}} \log q_\phi(\mathbf{y}|\mathbf{P})$

Figure 2: The training loop of A-NESI.

This factorization makes the inference model highly flexible. We can use simpler models if we know the dependencies between variables in $W$ and $Y$. The factorization is computationally linear in $k_W + k_Y$, instead of exponential in $k_W$ for exact inference. During testing, we use a beam search to find the most likely prediction from $q_{\phi,\psi}(\mathbf{w}, \mathbf{y}|\mathbf{P})$. If the method successfully trained the perception model, then its entropy is low and a beam search should easily find the most likely prediction [37].

There are no restrictions on the architecture of the inference model $q_\phi$. Any neural network with appropriate inductive biases and parameter sharing to speed up training can be chosen. For instance, CNNs over grids of variables, graph neural networks [27, 48] for reasoning problems on graphs, or transformers for sets or sequences [55].

We use the prediction model to train the perception model $f_{\boldsymbol{\theta}}$ given a dataset $\mathcal{D}$ of tuples $(\mathbf{x}, \mathbf{y})$. Our novel loss function trains the perception model by backpropagating through the prediction model:

$$\mathcal{L}_{Perc}(\mathcal{D}, \boldsymbol{\theta}) = -\log q_\phi(\mathbf{y}|\mathbf{P} = f_{\boldsymbol{\theta}}(\mathbf{x})), \quad \mathbf{x}, \mathbf{y} \sim \mathcal{D} \tag{4}$$

The gradients of this loss are biased due to the error in the approximation $q_\phi$, but it has no variance outside of sampling from the training dataset.

## 3.2 Training the inference model

We define two variations of our method. The *prediction-only* variant (Section 3.2.1) uses only the prediction model $q_\phi(\mathbf{y}|\mathbf{P})$, while the *explainable* variant (Section 3.2.4) also uses the explanation model $q_\psi(\mathbf{w}|\mathbf{y}, \mathbf{P})$.

Efficient training of the inference model relies on two design decisions. The first is a descriptive factorization of the output space $Y$, which we discuss in Section 3.2.2. The second is using an informative *belief prior* $p(\mathbf{P})$, which we discuss in Section 3.2.3.

We first define the forward process that uses the symbolic function $c$ to generate training data:

$$p(\mathbf{w}, \mathbf{y}|\mathbf{P}) = p(\mathbf{w}|\mathbf{P})p(\mathbf{y}|\mathbf{w}, \mathbf{P}) = p(\mathbf{w}|\mathbf{P})\mathbb{1}_{c(\mathbf{w})=\mathbf{y}} \tag{5}$$

We take some example world $\mathbf{w}$ and deterministically compute the output of the symbolic function $c(\mathbf{w})$. Then, we compute whether the output $c(\mathbf{w})$ equals $\mathbf{y}$. Therefore, $p(\mathbf{w}, \mathbf{y}|\mathbf{P})$ is 0 if $c(\mathbf{w}) \neq \mathbf{y}$ (that is, $\mathbf{w}$ is not a possible world of $\mathbf{y}$).

The belief prior $p(\mathbf{P})$ allows us to generate beliefs $\mathbf{P}$ for the forward process. That is, we generate training data for the inference model using

$$p(\mathbf{P}, \mathbf{w}) = p(\mathbf{P})p(\mathbf{w}|\mathbf{P})$$
$$\text{where } \mathbf{P}, \mathbf{w} \sim p(\mathbf{P}, \mathbf{w}), \quad \mathbf{y} = c(\mathbf{w}). \tag{6}$$

The belief prior allows us to train the inference model with synthetic data: The prior and the forward process define everything $q_\phi$ needs to learn. Note that the sampling process $\mathbf{P}, \mathbf{w} \sim p(\mathbf{P}, \mathbf{w})$ is fast and parallelisable. It involves sampling from a Dirichlet distribution, and then sampling from a categorical distribution for each dimension of $\mathbf{w}$ based on the sampled parameters $\mathbf{P}$.

### 3.2.1 Training the prediction-only variant

In the prediction-only variant, we only train the prediction model $q_\phi(\mathbf{y}|\mathbf{P})$. We use the samples generated by the process in Equation 6. We minimize the expected cross entropy between $p(\mathbf{y}|\mathbf{P})$

and $q_\phi(\mathbf{y}|\mathbf{P})$ over the prior $p(\mathbf{P})$:

$$\mathcal{L}_{Pred}(\phi) = -\log q_\phi(c(\mathbf{w})|\mathbf{P}), \quad \mathbf{P}, \mathbf{w} \sim p(\mathbf{P}, \mathbf{w}) \tag{7}$$

See A for a derivation. In the loss function defined in Equation 7, we estimate the expected cross entropy using samples from $p(\mathbf{P}, \mathbf{w})$. We use the sampled world $\mathbf{w}$ to compute the output $\mathbf{y} = c(\mathbf{w})$ and increase its probability under $q_\phi$. Importantly, we do not need to use any data to evaluate this loss function. We give pseudocode for the full training loop in Figure 2.

### 3.2.2 Output space factorization

The factorization of the output space $Y$ introduced in Section 2 is one of the key ideas that allow efficient learning in A-NESI. We will illustrate this with MNISTAdd. As the number of digits $N$ increases, the number of possible outputs (i.e., sums) is $2 \cdot 10^N - 1$. Without factorization, we would need an exponentially large output layer. We solve this by predicting the individual digits of the output so that we need only $N \cdot 10 + 2$ outputs similar to [3]. Furthermore, recognizing a single digit of the sum is easier than recognizing the entire sum: for its rightmost digit, only the rightmost digits of the input are relevant.

Choosing the right factorization is crucial when applying A-NESI. A general approach is to take the CNF of the symbolic function and predict each clause's truth value. However, this requires grounding the formula, which can be exponential. Another option is to predict for what objects a universally quantified formula holds, which would be linear in the number of objects.

### 3.2.3 Belief prior design

How should we choose the $\mathbf{P}$s for which we train $q_{\phi,\psi}$? A naive method would use the perception model $f_\theta$, sample some training data $\mathbf{x}_1, ..., \mathbf{x}_k \sim \mathcal{D}$ and train the inference model over $\mathbf{P}_1 = f_\theta(\mathbf{x}_1), ..., \mathbf{P}_k = f_\theta(\mathbf{x}_k)$. However, this means the inference model is only trained on those $\mathbf{P}$ occurring in the training data. Again, consider the Multi-digit MNISTAdd problem. For $N = 15$, we have a training set of 2000 sums, while there are $2 \cdot 10^{15} - 1$ possible sums. By simulating many beliefs, the inference model sees a much richer set of inputs and outputs, allowing it to generalize.

A better approach is to fit a Dirichlet prior $p(\mathbf{P})$ on $\mathbf{P}_1, ..., \mathbf{P}_k$ that covers all possible combinations of numbers. We choose a Dirichlet prior since it is conjugate to the discrete distributions. For details, see Appendix F. During hyperparameter tuning, we found that the prior needs to be high entropy to prevent the inference model from ignoring the inputs $\mathbf{P}$. Therefore, we regularize the prior with an additional term encouraging high-entropy Dirichlet distributions.

### 3.2.4 Training the explainable variant

The explainable variant uses both the prediction model $q_\phi(\mathbf{y}|\mathbf{P})$ and the optional explanation model $q_\psi(\mathbf{w}|\mathbf{y}, \mathbf{P})$. When training the explainable variant, we use the idea that both factorizations of the joint should have the same probability mass, that is, $p(\mathbf{w}, \mathbf{y}|\mathbf{P}) = q_{\phi,\psi}(\mathbf{w}, \mathbf{y}|\mathbf{P})$. To this end, we use a novel *joint matching* loss inspired by the theory of GFlowNets [9, 10], in particular, the trajectory balance loss introduced by [33] which is related to variational inference [34]. For an in-depth discussion, see Appendix E. The joint matching loss is essentially a regression of $q_{\phi,\psi}$ onto the true joint $p$ that we compute in closed form:

$$\mathcal{L}_{Expl}(\phi, \psi) = \left( \log \frac{q_{\phi,\psi}(\mathbf{w}, c(\mathbf{w})|\mathbf{P})}{p(\mathbf{w}|\mathbf{P})} \right)^2, \quad \mathbf{P}, \mathbf{w} \sim p(\mathbf{P}, \mathbf{w}) \tag{8}$$

Here we use that $p(\mathbf{w}, c(\mathbf{w})|\mathbf{P}) = p(\mathbf{w}|\mathbf{P})$ since $c(\mathbf{w})$ is deterministic. Like when training the prediction-only variant, we sample a random belief $\mathbf{P}$ and world $\mathbf{w}$ and compute the output $\mathbf{y}$. Then we minimize the loss function to *match* the joints $p$ and $q_{\phi,\psi}$. We further motivate the use of this loss in Appendix D. Instead of a classification loss like cross-entropy, the joint matching loss ensures $q_{\phi,\psi}$ does not become overly confident in a single prediction and allows spreading probability mass easier.

### 3.3 Symbolic pruner

An attractive option is to use symbolic knowledge to ensure the inference model only generates valid outputs. We can compute each factor $q_\psi(w_i|\mathbf{y}, \mathbf{w}_{1:i-1}, \mathbf{P})$ (both for world and output variables)

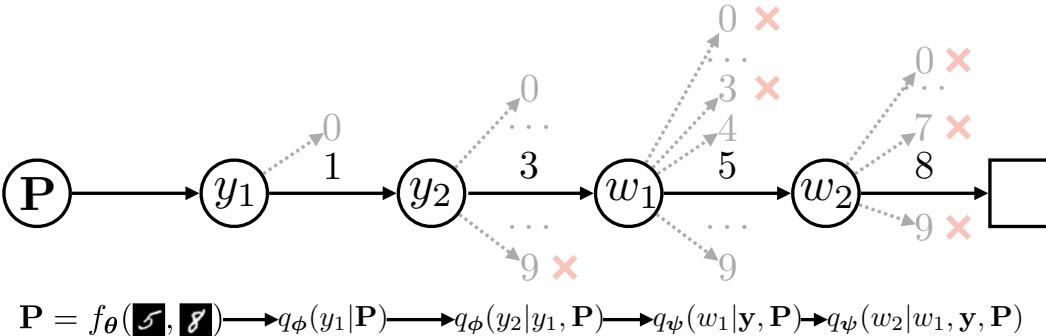

Figure 3: Example rollout sampling an explainable variant on input $\mathbf{x} = (\text{5}, \text{8})$. For $y_2$, we prune option 9, as the highest attainable sum is $9 + 9 = 18$. For $w_1$, $\{0, ..., 3\}$ are pruned as there is no second digit to complete the sum. $w_2$ is deterministic given $w_1$, and prunes all branches but 8.

using a neural network $\hat{q}_{\boldsymbol{\psi},i}$ and a *symbolic pruner* $s_{\mathbf{y},\mathbf{w}_{1:i-1}} : W_i \rightarrow \{0,1\}$:

$$q_{\boldsymbol{\psi}}(w_i = j|\mathbf{y}, \mathbf{w}_{1:i-1}, \mathbf{P}) = \frac{\hat{q}_{\boldsymbol{\psi}}(w_i = j|\mathbf{y}, \mathbf{w}_{1:i-1}, \mathbf{P})s_{\mathbf{y},\mathbf{w}_{1:i-1}}(j)}{\sum_{j' \in W_i} \hat{q}_{\boldsymbol{\psi}}(w_i = j'|\mathbf{y}, \mathbf{w}_{1:i-1}, \mathbf{P})s_{\mathbf{y},\mathbf{w}_{1:i-1}}(j')} \tag{9}$$

The symbolic pruner sets the probability mass of certain choices for the variable $w_i$ to zero. Then, $q_{\boldsymbol{\psi}}$ is computed by renormalizing. If we know that by expanding $\mathbf{w}_{1:i-1}$ it will be impossible to produce a possible world for $\mathbf{y}$, we can set the probability mass under that branch to 0: we will know that $p(\mathbf{w}, \mathbf{y}) = 0$ for all such branches. In Figure 3 we give an example for single-digit MNISTAdd. Symbolic pruning significantly reduces the number of branches our algorithm needs to explore during training. Moreover, symbolic pruning is critical in settings where verifiability and safety play crucial roles, such as medicine. We discuss the design of symbolic pruners $s$ in Appendix G.

## 4 Experiments

We study three Neurosymbolic reasoning tasks to evaluate the performance and scalability of A-NeSI: Multi-digit MNISTAdd [37], Visual Sudoku Puzzle Classification [4] and Warcraft path planning. Code is available at `https://github.com/HEmile/a-nesi`. We used the ADAM optimizer throughout.

A-NESI has two prediction methods. 1) **Symbolic prediction** uses the symbolic reasoning function $c$ to compute the output: $\hat{\mathbf{y}} = c(\arg\max_{\mathbf{w}} p(\mathbf{w}|\mathbf{P} = f_{\boldsymbol{\theta}}(\mathbf{x})))$. 2) **Neural prediction** predicts with the prediction network $q_{\boldsymbol{\phi}}$ using a beam search: $\hat{\mathbf{y}} = \arg\max_{\mathbf{y}} q_{\boldsymbol{\phi}}(\mathbf{y}|\mathbf{P} = f_{\boldsymbol{\theta}}(\mathbf{x}))$. In our studied tasks, neural prediction cannot perform better than symbolic prediction. However, we still use the prediction model to efficiently train the perception model in the symbolic prediction setting. We consider the prediction network adequately trained if it matches the accuracy of symbolic prediction.

### 4.1 Multi-digit MNISTAdd

We evaluate A-NESI on the Multi-Digit MNISTAdd task (Section 2). For the perception model, we use the same CNN as in DeepProbLog [35]. The prediction model has $N+1$ factors $q_{\boldsymbol{\phi}}(y_i|\mathbf{y}_{1:i-1}, \mathbf{P})$, while the explanation model has $2N$ factors $q_{\boldsymbol{\psi}}(w_i|\mathbf{y}, \mathbf{w}_{1,i-1}, \mathbf{P})$. We model each factor with a separate MLP. $y_i$ and $w_i$ are one-hot encoded digits, except for the first output digit $y_1$: it can only be 0 or 1. We used a shared set of hyperparameters for all $N$. For more details and a description of the baselines, see Appendix I.1.

Table 1 reports the accuracy of predicting the sum. For all $N$, A-NESI is close to the reference accuracy, meaning there is no significant drop in the accuracy of the perception model as $N$ increases. For small $N$, it is slightly outperformed by DeepStochLog [56], which can not scale to $N = 15$. A-NESI also performs slightly better than DeepProbLog, showing approximate inference does not hurt the accuracy. With neural prediction, we get the same accuracy for low $N$, but there is a significant drop for $N = 15$, meaning the prediction network did not perfectly learn the problem. However, compared to training a neural network without background knowledge (Embed2Sym with

| | N=1 | N=2 | N=4 | N=15 |
|---|---|---|---|---|
| | **Symbolic prediction** | | | |
| LTN | $80.54 \pm 23.33$ | $77.54 \pm 35.55$ | T/O | T/O |
| DeepProbLog | $97.20 \pm 0.50$ | $95.20 \pm 1.70$ | T/O | T/O |
| DPLA* | $88.90 \pm 14.80$ | $83.60 \pm 23.70$ | T/O | T/O |
| DeepStochLog | $\mathbf{97.90 \pm 0.10}$ | $\mathbf{96.40 \pm 0.10}$ | $\mathbf{92.70 \pm 0.60}$ | T/O |
| Embed2Sym | $97.62 \pm 0.29$ | $93.81 \pm 1.37$ | $91.65 \pm 0.57$ | $60.46 \pm 20.36$ |
| A-NESI (predict) | $97.66 \pm 0.21$ | $95.96 \pm 0.38$ | $92.56 \pm 0.79$ | $75.90 \pm 2.21$ |
| A-NESI (explain) | $97.37 \pm 0.32$ | $96.04 \pm 0.46$ | $92.11 \pm 1.06$ | $\mathbf{76.84 \pm 2.82}$ |
| A-NESI (pruning) | $97.57 \pm 0.27$ | $95.82 \pm 0.33$ | $92.40 \pm 0.68$ | $76.46 \pm 1.39$ |
| A-NESI (no prior) | $76.54 \pm 27.38$ | $95.67 \pm 0.53$ | $44.58 \pm 38.34$ | $0.03 \pm 0.09$ |
| | **Neural prediction** | | | |
| Embed2Sym | $97.34 \pm 0.19$ | $84.35 \pm 6.16$ | $0.81 \pm 0.12$ | $0.00$ |
| A-NESI (predict) | $\mathbf{97.66 \pm 0.21}$ | $95.95 \pm 0.38$ | $\mathbf{92.48 \pm 0.76}$ | $54.66 \pm 1.87$ |
| A-NESI (explain) | $97.37 \pm 0.32$ | $\mathbf{96.05 \pm 0.47}$ | $92.14 \pm 1.05$ | $\mathbf{61.77 \pm 2.37}$ |
| A-NESI (pruning) | $97.57 \pm 0.27$ | $95.82 \pm 0.33$ | $92.38 \pm 0.64$ | $59.88 \pm 2.95$ |
| A-NESI (no prior) | $76.54 \pm 27.01$ | $95.28 \pm 0.62$ | $40.76 \pm 34.29$ | $0.00 \pm 0.00$ |
| Reference | $98.01$ | $96.06$ | $92.27$ | $73.97$ |

Table 1: Test accuracy of predicting the correct sum on the Multi-digit MNISTAdd task. "T/O" (timeout) represent computational timeouts. Reference accuracy approximates the accuracy of an MNIST predictor with $0.99$ accuracy using $0.99^{2N}$. Bold numbers signify the highest average accuracy for some $N$ within the prediction categories. *predict* is the prediction-only variant, *explain* is the explainable variant, *pruning* adds symbolic pruning (see Appendix H), and *no prior* is the prediction-only variant *without* the prior $p(\mathbf{P})$.

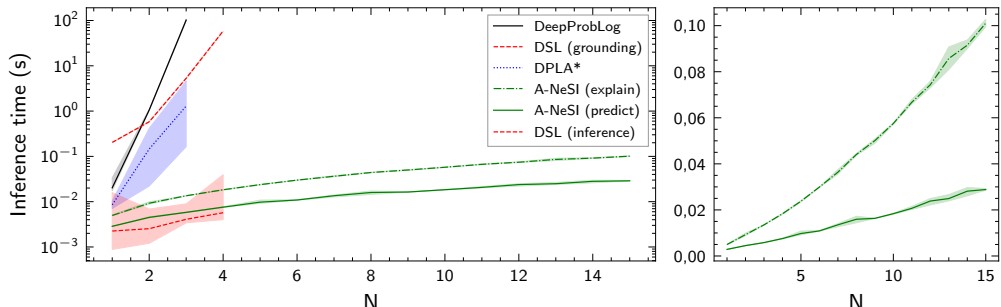

Figure 4: Inference time for a single input $\mathbf{x}$ for different nr. of digits. The left plot uses a log scale, and the right plot a linear scale. DSL stands for DeepStochLog. We use the GM variant of DPLA*.

neural prediction), it is much more accurate already for $N = 2$. Therefore, A-NESI's training loop allows training a prediction network with high accuracy on large-scale problems.

Comparing the different A-NESI variants, we see the prediction-only, explainable and pruned variants perform quite similarly, with significant differences only appearing at $N = 15$ where the explainable and pruning variants outperform the predict-only model, especially on neural prediction. However, when removing the prior $p(\mathbf{P})$, the performance degrades quickly. The prediction model sees much fewer beliefs $\mathbf{P}$ than when sampling from a (high-entropy) prior $p(\mathbf{P})$. A second and more subtle reason is that at the beginning of training, all the beliefs $\mathbf{P} = f_{\boldsymbol{\theta}}(\mathbf{x})$ will be uniform because the perception model is not yet trained. Then, the prediction model learns to ignore the input belief $\mathbf{P}$.

Figure 4 shows the runtime for inference on a single input $\mathbf{x}$. Inference in DeepProbLog (corresponding to exact inference) increases with a factor 100 as $N$ increases, and DPLA* (another approximation) is not far behind. Inference in DeepStochLog, which uses different semantics, is efficient due to caching but requires a grounding step that is exponential both in time and memory. We could not ground beyond $N = 4$ because of memory issues. A-NESI avoids having to perform grounding altogether: it scales slightly slower than linear. Furthermore, it is much faster in practice as parallelizing the computation of multiple queries on GPUs is trivial.

|  | N=4 | N=9 |
|---|---|---|
| CNN | $51.50 \pm 3.34$ | $51.20 \pm 2.20$ |
| Exact inference | $86.70 \pm 0.50$ | T/O |
| NeuPSL | $\mathbf{89.7 \pm 2.20}$ | $51.50 \pm 1.37$ |
| A-NeSI (symbolic prediction) | $\mathbf{89.70 \pm 2.08}$ | $\mathbf{62.15 \pm 2.08}$ |
| A-NeSI (neural prediction) | $\mathbf{89.80 \pm 2.11}$ | $\mathbf{62.25 \pm 2.20}$ |

Table 2: Test accuracy of predicting whether a grid of numbers is a Sudoku. For A-NeSI, we used the prediction-only variant.

## 4.2 Visual Sudoku Puzzle Classification

In this task, the goal is to recognize whether an $N \times N$ grid of MNIST digits is a Sudoku. We have 100 examples of correct and incorrect grids from [4]. We use the same MNIST classifier as in the previous section. We treat $\mathbf{w}$ as a grid of digits in $\{1, \ldots, N\}$ and designed an easy-to-learn representation of the label. For sudokus, all pairs of digits $\mathbf{w}_i, \mathbf{w}_j$ in a row, column, and block must differ. For each such pair, a dimension in $\mathbf{y}$ is 1 if different and 0 otherwise, and the symbolic function $c$ computes these interactions from $\mathbf{w}$. The prediction model is a single MLP that takes the probabilities for the digits $\mathbf{P}_i$ and $\mathbf{P}_j$ and predicts the probability that those represent different digits. For additional details, see Appendix I.2.

Table 2 shows the accuracy of classifying Sudoku puzzles. A-NeSI is the only method to perform better than random on $9 \times 9$ sudokus. Exact inference cannot scale to $9 \times 9$ sudokus, while we were able to run A-NeSI for 3000 epochs in 38 minutes on a single NVIDIA RTX A4000.

## 4.3 Warcraft Visual Path Planning

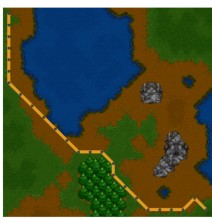

Figure 5: Example of the Warcraft task.

|  | $12 \times 12$ | $30 \times 30$ |
|---|---|---|
| ResNet18 [43] | $23.0 \pm 0.3$ | $0.0 \pm 0.0$ |
| SPL [1] | 78.2 | T/O |
| I-MLE [41] | $97.2 \pm 0.5$ | $\mathbf{93.7 \pm 0.6}$ |
| RLOO [29] | $43.75 \pm 12.35$ | $12.59 \pm 16.38$ |
| A-NeSI | $94.57 \pm 2.27$ | $17.13 \pm 16.32$ |
| A-NeSI + RLOO | $\mathbf{98.96 \pm 1.33}$ | $67.57 \pm 36.76$ |

Table 3: Test accuracy of predicting the lowest-cost path on the Warcraft Visual Path Planning task. For A-NeSI, we used the prediction-only variant.

The Warcraft Visual Path Planning task [43] is to predict a shortest path from the top left corner of an $N \times N$ grid to the bottom right, given an image from the Warcraft game (Figure 5). The perception model has to learn the cost of each tile of the Warcraft image. We use a small CNN that takes the tile $i, j$ (a $3 \times 8 \times 8$ image) and outputs a distribution over 5 possible costs. The symbolic function $c(\mathbf{w})$ is Dijkstra's algorithm and returns a shortest path $\mathbf{y}$, which we encode as a sequence of the 8 (inter-)cardinal directions starting from the top-left corner. We use a ResNet18 [22] as the prediction model, which we train to predict the next direction given beliefs of the tile costs $\mathbf{P}$ and the current location. We pretrain the prediction model on a fixed prior and train the perception model with a frozen prediction model. See Appendix I.3 for additional details.

Table 3 presents the accuracy of predicting a shortest path. A-NeSI is competitive on $12 \times 12$ grids but struggles on $30 \times 30$ grids. We believe this is because the prediction model is not accurate enough, resulting in gradients with too high bias. Still, A-NeSI finds short paths, is far better than a pure neural network, and can scale to $30 \times 30$ grids, unlike SPL [1], which uses exact inference. Since both SPL and I-MLE [41] have significantly different setups (see Appendix I.3.2), we added experiments using REINFORCE with the leave-one-out baseline (RLOO, [29]) that we implemented with the Storchastic PyTorch library [53]. We find that RLOO has high variance in its performance. Since RLOO is unbiased with high variance and A-NeSI is biased with no variance, we also tried running A-NeSI and RLOO simultaneously. Interestingly, this is the best-performing method on the $12 \times 12$ grid, and has competitive performance on $30 \times 30$ grids, albeit with high variance (6 out 10 runs get to an accuracy between 93.3% and 98.5%, while the other runs are stuck around 25% accuracy).

# 5  Related work

A-NESI can approximate multiple PNL methods [16]. DeepProbLog [35] performs symbolic reasoning by representing $\mathbf{w}$ as ground facts in a Prolog program. It enumerates all possible proofs of a query $\mathbf{y}$ and weights each proof by $p(\mathbf{w}|\mathbf{P})$. NeurASP [58] is a PNL framework closely related to DeepProbLog, but is based on Answer Set Programming [12]. Some methods consider constrained structured output prediction [19]. In Appendix B, we discuss extending A-NESI to this setting. Semantic Loss [57] improves learning with a loss function but does not guarantee that formulas are satisfied at test time. Like A-NESI, Semantic Probabilistic Layers [1] solves this with a layer that performs constrained prediction. These approaches perform exact inference using probabilistic circuits (PCs) [60]. Other methods perform approximate inference by only considering the top-k proofs in PCs [36, 23]. However, finding those proofs is hard, especially when beliefs have high entropy, and limiting to top-k significantly reduces performance. Other work considers MCMC approximations [31]. Using neural networks for approximate inference ensures computation time is constant and independent of the entropy of $\mathbf{P}$ or long MCMC chains.

Other neurosymbolic methods use fuzzy logics [5, 17, 15, 18], which are faster than PNL with exact inference. Although traversing ground formulas is linear time, the grounding is itself often exponential [37], so the scalability of fuzzy logics often fails to deliver. A-NESI is polynomial in the number of ground atoms and does not traverse the ground formula. Furthermore, background knowledge is often not fuzzy [54, 20], and fuzzy semantics does not preserve classical equivalence.

A-NESI performs gradient estimation of the WMC problem. We can extend our method to biased but zero-variance gradient estimation by learning a distribution over function outputs (see Appendix C). Many recent works consider continuous relaxations of discrete computation to make them differentiable [42, 24] but require many tricks to be computationally feasible. Other methods compute MAP states to compute the gradients [41, 11, 47] but are restricted to integer linear programs. The score function (or 'REINFORCE') gives unbiased yet high-variance gradient estimates [40, 53]. Variance reduction techniques, such as memory augmentation [32] and leave-one-out baselines [29], exist to reduce this variance.

# 6  Conclusion, Discussion and Limitations

We introduced A-NESI, a scalable approximate method for probabilistic neurosymbolic learning. We demonstrated that A-NESI scales to combinatorially challenging tasks without losing accuracy. A-NESI can be extended to include explanations and hard constraints without loss of performance.

However, there is no 'free lunch': when is A-NESI a good approximation? We discuss three aspects of learning tasks that could make it difficult to learn a strong and efficient inference model.

- **Dependencies of variables.** When variables in world $\mathbf{w}$ are highly dependent, finding an informative prior is hard. We suggest then using a prior that can incorporate dependencies such as a normalizing flow [45, 14] or deep generative models [51] over a Dirichlet distribution.

- **Structure in symbolic reasoning function.** We studied reasoning tasks with a relatively simple structure. Learning the inference model will be more difficult when the symbolic function $c$ is less structured. Studying the relation between structure and learnability is interesting future work.

- **Problem size.** A-NESI did not perfectly train the prediction model in more challenging problems, which is evident from the divergence in performance between symbolic and neural prediction for 15 digits in Table 1. We expect its required parameter size and training time to increase with the problem size.

Promising future avenues are studying if the explanation model produces helpful explanations [50], extensions to continuous random variables [21] (see Appendix C for an example), and extensions to unnormalized distributions such as Markov Logic Networks[46], as well as (semi-) automated A-NESI solutions for neurosymbolic programming languages like DeepProbLog [35].

## Acknowledgements

We want to thank Robin Manhaeve, Giusseppe Marra, Thomas Winters, Yaniv Aspis, Robin Smet, and Anna Kuzina for helpful discussions. For our experiments, we used the DAS-6 compute cluster [6]. We also thank SURF (www.surf.nl) for its support in using the Lisa Compute Cluster.

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

## A Derivation of prediction-only variant loss

$$\mathbb{E}_{p(\mathbf{P})}\left[-\mathbb{E}_{p(\mathbf{y}|\mathbf{P})}[\log q_\phi(\mathbf{y}|\mathbf{P})]\right] \tag{10}$$

$$= -\mathbb{E}_{p(\mathbf{P})}\left[\mathbb{E}_{p(\mathbf{w},\mathbf{y}|\mathbf{P})}[\log q_\phi(\mathbf{y}|\mathbf{P})]\right] \tag{11}$$

$$= -\mathbb{E}_{p(\mathbf{P},\mathbf{w})}\left[\log q_\phi(c(\mathbf{w})|\mathbf{P})\right] \tag{12}$$

$$\mathcal{L}_{Pred}(\phi) = -\log q_\phi(c(\mathbf{w})|\mathbf{P}), \quad \mathbf{P},\mathbf{w} \sim p(\mathbf{P},\mathbf{w}) \tag{13}$$

In line 11, we marginalize out $\mathbf{w}$, and use the fact that $\mathbf{y}$ is deterministic given $\mathbf{w}$.

## B Constrained structured output prediction

We consider how to implement the constrained structured output prediction task considered in (for example) [2, 1, 57] in the A-NESI framework. Here, the goal is to learn a mapping of some $X$ to a structured *output* space $W$, where we have some constraint $c(\mathbf{w})$ that returns 1 if the background knowledge holds, and 0 otherwise. We can model the constraints using $Y = \{0, 1\}$; that is, the 'output' in our problem setup is whether $\mathbf{w}$ satisfies the background knowledge $c$ or not. We give an example of this setting in Figure 6.

Then, we design the inference model as follows. 1) $q_\phi(y|\mathbf{P})$ is tasked with predicting the probability that randomly sampled outputs $\mathbf{w} \sim p(\mathbf{w}|\mathbf{P})$ will satisfy the background knowledge. 2) $q_\psi(\mathbf{w}|y = 1, \mathbf{P})$ is an approximate posterior over structured outputs $\mathbf{w}$ that satisfy the background knowledge $c$.

This setting changes the interpretation of the set $W$ from *unobserved* worlds to *observed* outputs. We will train our perception module using a "strongly" supervised learning loss where $\mathbf{x}, \mathbf{w} \sim \mathcal{D}_\mathcal{L}$:

$$\mathcal{L}_{Perc}(\boldsymbol{\theta}) = -\log q_\psi(\mathbf{w}|y = 1, \mathbf{P} = f_{\boldsymbol{\theta}}(\mathbf{x})). \tag{14}$$

If we also have unlabeled data $\mathcal{D}_U$, we can use the prediction model to ensure the perception model gives high probabilities for worlds that satisfy the background knowledge. This approximates Semantic Loss [57]: Given $\mathbf{x} \sim \mathcal{D}_U$,

$$\mathcal{L}_{SL}(\boldsymbol{\theta}) = -\log q_\phi(y = 1|\mathbf{P} = f_{\boldsymbol{\theta}}(\mathbf{x})). \tag{15}$$

That is, we have some input $\mathbf{x}$ for which we have no labelled output. Then, we increase the probability that the belief $\mathbf{P}$ the perception module $f_\theta$ predicts for $\mathbf{x}$ would sample structured outputs $\mathbf{w}$ that satisfy the background knowledge.

Training the inference model in this setting can be challenging if the problem is very constrained. Then, random samples $\mathbf{P}, \mathbf{w} \sim p(\mathbf{P}, \mathbf{w})$ will usually not satisfy the background knowledge. Since we are only in the case that $y = 1$, we can choose to sample from the inference model $q_\phi$ and exploit the symbolic pruner to obtain samples that are guaranteed to satisfy the background knowledge. Therefore, we modify equation 8 to the *on-policy joint matching loss*

$$\mathcal{L}_{Expl}(\mathbf{P}, \phi, \psi) = \mathbb{E}_{q_\psi(\mathbf{w}|y=1,\mathbf{P})}\left[\left(\log \frac{q_{\phi,\psi}(\mathbf{w}, y = 1|\mathbf{P})}{p(\mathbf{w}|\mathbf{P})}\right)^2\right] \tag{16}$$

Here, we incur some sampling bias by not sampling structured outputs from the true posterior, but this bias will reduce as $q_\phi$ becomes more accurate. We can also choose to combine the on- and off-policy losses. Another option to make learning easier is using the suggestions of Section 3.2.2: factorize $y$ to make it more fine-grained.

## C A-NESI as a Gradient Estimation method

In this appendix, we discuss using the method A-NESI introduced for general gradient estimation [40]. We first define the gradient estimation problem. Consider some neural network $f_{\boldsymbol{\theta}}$ that predicts the parameters $\mathbf{P}$ of a distribution over unobserved variable $\mathbf{z} \in Z$: $p(\mathbf{z}|\mathbf{P} = f_{\boldsymbol{\theta}}(\mathbf{x}))$. This distribution corresponds to the distribution over worlds $p(\mathbf{w}|\mathbf{P})$ in A-NESI. Additionally, assume we have some deterministic function $g(\mathbf{z})$ that we want to maximize in expectation. This maximization requires estimating the gradient

$$\nabla_{\boldsymbol{\theta}} \mathbb{E}_{p(\mathbf{z}|\mathbf{P}=f_{\boldsymbol{\theta}}(\mathbf{x}))}[g(\mathbf{z})]. \tag{17}$$

Common methods for estimating this gradient are reparameterization [26], which only applies to continuous random variables and differentiable $r$, and the score function [40, 49] which has notoriously high variance.

Instead, our gradient estimation method learns an *inference model* $q_\phi(r|\mathbf{P})$ to approximate the distribution of outcomes $r = g(\mathbf{z})$ for a given $\mathbf{P}$. In A-NESI, this is the prediction network $q_\phi(\mathbf{y}|\mathbf{P})$ that estimates the WMC problem of Equation 1. Approximating a *distribution* over outcomes is similar to the idea of distributional reinforcement learning [8]. Our approach is general: Unlike reparameterization, we can use inference models in settings with discrete random variables $\mathbf{z}$ and non-differentiable downstream functions $g$.

We derive the training loss for our inference model similar to that in Section 3.2.1. First, we define the joint on latents $\mathbf{z}$ and outcomes $r$ like the joint process in 6 as $p(r, \mathbf{z}|\mathbf{P}) = p(\mathbf{z}|\mathbf{P}) \cdot \delta_{g(z)}(r)$, where $\delta_{g(z)}(r)$ is the dirac-delta distribution that checks if the output of $g$ on $\mathbf{z}$ is equal to $r$. Then we introduce a prior over distribution parameters $p(\mathbf{P})$, much like the prior over beliefs in A-NESI. An obvious choice is to use a prior conjugate to $p(\mathbf{z}|\mathbf{P})$. We minimize the expected KL-divergence between $p(r|\mathbf{P})$ and $q_\phi(r|\mathbf{P})$:

$$\mathbb{E}_{p(\mathbf{P})}[D_{KL}(p||q_\phi)] \tag{18}$$

$$= \mathbb{E}_{p(\mathbf{P})}\left[\mathbb{E}_{p(r|\mathbf{P})}[\log q_\phi(r|\mathbf{P})]\right] + C \tag{19}$$

$$= \mathbb{E}_{p(\mathbf{P})}\left[\int_{\mathbb{R}} p(r|\mathbf{P}) \log q_\phi(r|\mathbf{P})]dr\right] + C \tag{20}$$

Next, we marginalize over $\mathbf{z}$, dropping the constant:

$$\mathbb{E}_{p(\mathbf{P})}\left[\int_Z \int_{\mathbb{R}} p(r, \mathbf{z}|\mathbf{P}) \log q_\phi(r|\mathbf{P})]drd\mathbf{z}\right] \tag{21}$$

$$= \mathbb{E}_{p(\mathbf{P})}\left[\int_Z p(\mathbf{z}|\mathbf{P}) \int_{\mathbb{R}} \delta_{g(\mathbf{z})}(r) \log q_\phi(r|\mathbf{P})]drd\mathbf{z}\right] \tag{22}$$

$$= \mathbb{E}_{p(\mathbf{P})}\left[\int_Z p(\mathbf{z}|\mathbf{P}) \log q_\phi(g(\mathbf{z})|\mathbf{P})]d\mathbf{z}\right] \tag{23}$$

$$= \mathbb{E}_{p(\mathbf{P},\mathbf{z})}[\log q_\phi(g(\mathbf{z})|\mathbf{P})] \tag{24}$$

This gives us a negative-log-likelihood loss function similar to Equation 7.

$$\mathcal{L}_{Inf}(\phi) = -\log q_\phi(g(\mathbf{z})|\mathbf{P}), \quad \mathbf{P}, \mathbf{z} \sim p(\mathbf{z}, \mathbf{P}) \tag{25}$$

where we sample from the joint $p(\mathbf{z}, \mathbf{P}) = p(\mathbf{P})p(\mathbf{z}|\mathbf{P})$.

We use a trained inference model to get gradient estimates:

$$\nabla_\mathbf{P} \mathbb{E}_{p(\mathbf{z}|\mathbf{P})}[g(\mathbf{z})] \approx \nabla_\mathbf{P} \mathbb{E}_{q_\phi(r|\mathbf{P})}[r] \tag{26}$$

We use the chain rule to update the parameters $\boldsymbol{\theta}$. This requires a choice of distribution $q_\phi(r|\mathbf{P})$ for which computing the mean $\mathbb{E}_{q_\phi(r|\mathbf{P})}[r]$ is easy. The simplest option is to parameterize $q_\phi$ with a univariate normal distribution. We predict the mean and variance using a neural network with parameters $\phi$. For example, a neural network $m_\phi$ would compute $\mu = m_\phi(\mathbf{P})$. Then, the mean parameter is the expectation on the right-hand side of Equation 26. The loss function for $f_\theta$ with this parameterization is:

$$\mathcal{L}_{NN}(\boldsymbol{\theta}) = -m_\phi(f_\theta(\mathbf{x})), \quad \mathbf{x} \sim \mathcal{D} \tag{27}$$

Interestingly, like A-NESI, this gives zero-variance gradient estimates! Of course, bias comes from the error in the approximation of $q_\phi$.

Like A-NESI, we expect the success of this method to rely on the ease of finding a suitable prior over $\mathbf{P}$ to allow proper training of the inference model. See the discussion in Section 3.2.3. We also expect that, like in A-NESI, it will be easier to train the inference model if the output $r = g(\mathbf{z})$ is structured instead of a single scalar. We refer to Section 3.2.2 for this idea. Other challenges might be stochastic and noisy output measurements of $r$ and non-stationarity of $g$, for instance, when training a VAE [26].

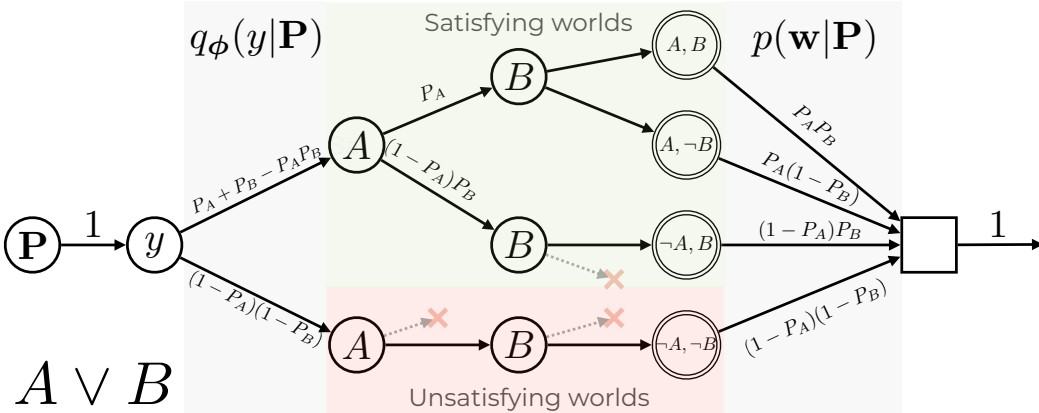

Figure 6: The tree flow network corresponding to weighted model counting on the formula $A \vee B$. Following edges upwards means setting the corresponding binary variable to true (and to false by following edges downwards). We first choose probabilities for the propositions $A$ and $B$, then choose whether we want to sample a world that satisfies the formula $A \vee B$. $y = 1$ is the WMC of $A \vee B$, and equals its outgoing flow $P_A + P_B - P_A P_B$. Terminal states (with two circles) represent choices of the binary variables $A$ and $B$. These are connected to a final sink node, corresponding to the prior over worlds $p(\mathbf{w}|\mathbf{P})$. The total ingoing and outgoing flow to this network is 1, as we deal with normalized probability distributions $p$ and $q_\phi$.

## D    A-NESI and GFlowNets

A-NeSI is heavily inspired by the theory of GFlowNets [9, 10], and we use this theory to derive our loss function. In the current section, we discuss these connections and the potential for future research by taking inspiration from the GFlowNet literature. In this section, we will assume the reader is familiar with the notation introduced in [10] and refer to this paper for the relevant background.

### D.1    Tree GFlowNet representation

The main intuition is that we can treat the inference model $q_\phi$ in Equation 3 as a 'trivial' GFlowNet. We refer to Figure 6 for an intuitive example. It shows what a flow network would look like for the formula $A \vee B$. We take the reward function $R(\mathbf{w}, \mathbf{y}) = p(\mathbf{w}, \mathbf{y})$. We represent states $s$ by $s = (\mathbf{P}, \mathbf{y}_{1:i}, \mathbf{w}_{1:j})$, that is, the belief $\mathbf{P}$, a list of some dimensions of the output instantiated with a value and a list of some dimensions of the world assigned to some value. Actions $a$ set some value to the next output or world variable, i.e., $A(s) = Y_{i+1}$ or $A(s) = W_{j+1}$.

Note that this corresponds to a flow network that is a tree everywhere but at the sink since the state representation conditions on the whole trajectory observed so far. We demonstrate this in Figure 6. We assume there is some fixed ordering on the different variables in the world, which we generate the value of one by one. Given this setup, Figure 6 shows that the branch going up from the node $y$ corresponds to the regular weighted model count (WMC) introduced in Equation 1.

The GFlowNet forward distribution $P_F$ is $q_\phi$ as defined in Equation 3. The backward distribution $P_B$ is $p(\mathbf{w}, \mathbf{y}|\mathbf{P})$ at the sink node, which chooses a terminal node. Then, since we have a tree, this determines the complete trajectory from the terminal node to the source node. Thus, at all other states, the backward distribution is deterministic. Since our reward function $R(\mathbf{w}, \mathbf{y}, \mathbf{P}) = p(\mathbf{w}, \mathbf{y}|\mathbf{P})$ is normalized, we trivially know the partition function $Z(\mathbf{P}) = \sum_{\mathbf{w}} \sum_{\mathbf{y}} R(\mathbf{w}, \mathbf{y}|\mathbf{P}) = 1$.

### D.2    Lattice GFlowNet representation

Our setup of the generative process assumes we are generating each variable in the world in some order. This is fine for some problems like MNISTAdd, where we can see the generative process as 'reading left to right'. For other problems, such as Sudoku, the order in which we would like to generate the symbols is less obvious. Would we generate block by block? Row by row? Column by column? Or is the assumption that it needs to be generated in some fixed order flawed by itself?

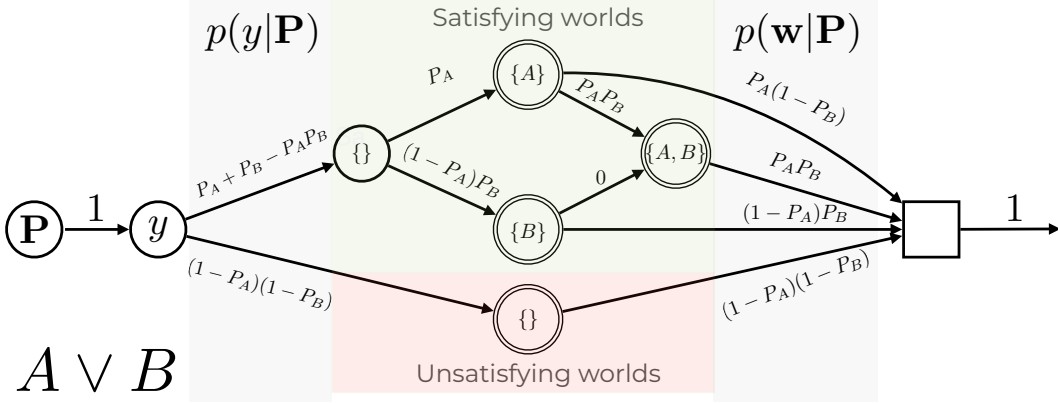

Figure 7: The lattice flow network corresponding to weighted model counting on the formula $A \vee B$. In this representation, nodes represent sets of included propositions. Terminal states represent sets of random variables such that $A \vee B$ is true given $y = 1$, or false otherwise.

In this section, we consider a second GFlowNet representation for the inference model that represents states using sets instead of lists. We again refer to Figure 7 for the resulting flow network of this representation for $A \vee B$. We represent states using $s = (\mathbf{P}, \{y_i\}_{i \in I_Y}, \{w_i\}_{i \in I_W})$, where $I_Y \subseteq \{1, ..., k_Y\}$ and $I_W \subseteq \{1, ..., k_W\}$ denote the set of variables for which a value is chosen. The possible actions from some state correspond to $A(s) = \bigcup_{i \notin I_W} W_i$ (and analogous for when $\mathbf{y}$ is not yet completely generated). For each variable in $W$ for which we do not have the value yet, we add its possible values to the action space.

With this representation, the resulting flow network is no longer a tree but a DAG, as the order in which we generate the different variables is now different for every trajectory. What do we gain from this? When we are primarily dealing with categorical variables, the two gains are 1) we no longer need to impose an ordering on the generative process, and 2) it might be easier to implement parameter sharing in the neural network that predicts the forward distributions, as we only need a single set encoder that can be reused throughout the generative process.

However, the main gain of the set-based approach is when worlds are all (or mostly) binary random variables. We illustrate this in Figure 7. Assume $W = \{0, 1\}^{k_W}$. Then we can have the following state and action representations: $s = (\mathbf{P}, \mathbf{y}, I_W)$, where $I_W \subseteq \{1, ..., k_W\}$ and $A(s) = \{1, ..., k_W\} \setminus I_W$. The intuition is that $I_W$ contains the set of all binary random variables that are set to 1 (i.e., true), and $\{1, ..., k_W\} \setminus I_W$ is the set of variables set to 0 (i.e., false). The resulting flow network represents a *partial order* over the set of all subsets of $\{1, ..., k_W\}$, which is a *lattice*, hence the name of this representation.

With this representation, we can significantly reduce the size and computation of the flow network required to express the WMC problem. As an example, compare Figures 6 and 7, which both represent the WMC of the formula $A \vee B$. We no longer need two nodes in the branch $y = 0$ to represent that we generate $A$ and $B$ to be false, as the initial empty set $\{\}$ already implies they are. This will save us two nodes. Similarly, we can immediately stop generating at $\{A\}$ and $\{B\}$, and no longer need to generate the other variable as false, which also saves a computation step.

While this is theoretically appealing, the three main downsides are 1) $P_B$ is no longer trivial to compute; 2) we have to handle the fact that we no longer have a tree, meaning there is no longer a unique optimal $P_F$ and $P_B$; and 3) parallelization becomes much trickier. We leave exploring this direction in practice for future work.

# E    Analyzing the Joint Matching Loss

This section discusses the loss function we use to train the joint variant in Equation 8. We recommend interested readers first read Appendix D.1. Throughout this section, we will refer to $p := p(\mathbf{w}, \mathbf{y}|\mathbf{P})$ (Equation 5) and $q := q_{\phi, \psi}(\mathbf{w}, \mathbf{y}|\mathbf{P})$ (Equation 2). We again refer to [10] for notational background.

### E.1 Trajectory Balance

We derive our loss function from the recently introduced Trajectory Balance loss for GFlowNets, which is proven to approximate the true Markovian flow when minimized. This means sampling from the GFlowNet allows sampling in proportion to reward $R(s_n) = p$. The Trajectory Balance loss is given by

$$\mathcal{L}(\tau) = \left( \log \frac{F(s_0) \prod_{t=1}^n P_F(s_t|s_{t-1})}{R(s_n) \prod_{t=1}^n P_B(s_{t-1}|s_t)} \right)^2 , \tag{28}$$

where $s_0$ is the source state, in our case $\mathbf{P}$, and $s_n$ is some terminal state that represents a full generation of $\mathbf{y}$ and $\mathbf{w}$. In the tree representation of GFlowNets for inference models (see Appendix D.1), this computation becomes quite simple:

1. $F(s_0) = 1$, as $R(s_n) = p$ is normalized;
2. $\prod_{t=1}^n P_F(s_t|s_{t-1}) = q$: The forward distribution corresponds to the inference model $q_\phi(\mathbf{w}, \mathbf{y}|\mathbf{P})$;
3. $R(s_n) = p$, as we define the reward to be the true joint probability distribution $p(\mathbf{w}, \mathbf{y}|\mathbf{P})$;
4. $\prod_{t=1}^n P_B(s_{t-1}|s_t) = 1$, since the backward distribution is deterministic in a tree.

Therefore, the trajectory balance loss for (tree) inference models is

$$\mathcal{L}(\mathbf{P}, \mathbf{y}, \mathbf{w}) = \left( \log \frac{q}{p} \right)^2 = (\log q - \log p)^2 , \tag{29}$$

i.e., the term inside the expectation of the joint matching loss in Equation 8. This loss function is stable because we can sum the individual probabilities in log-space.

A second question might then be how we obtain 'trajectories' $\tau = (\mathbf{P}, \mathbf{y}, \mathbf{w})$ to minimize this loss over. The paper on trajectory balance [33] picks $\tau$ *on-policy*, that is, it samples $\tau$ from the forward distribution (in our case, the inference model $q_{\phi,\psi}$). We discussed when this might be favorable in our setting in Appendix B (Equation 16). However, the joint matching loss as defined in Equation 8 is *off-policy*, as we sample from $p$ and not from $q_{\phi,\psi}$.

### E.2 Relation to common divergences

These questions open quite some design space, as was recently noted when comparing the trajectory balance loss to divergences commonly used in variational inference [34]. Redefining $P_F = q$ and $P_B = p$, the authors compare the trajectory balance loss with the KL-divergence and the reverse KL-divergence and prove that

$$\nabla_\phi D_{KL}(q||p) = \frac{1}{2} \mathbb{E}_{\tau \sim q}[\nabla_\phi \mathcal{L}(\tau)]. \tag{30}$$

That is, the *on*-policy objective minimizes the *reverse* KL-divergence between $p$ and $q$. We do not quite find such a result for the *off*-policy version we use for the joint matching loss in Equation 8:

$$\nabla_\phi D_{KL}(p||q) = -\mathbb{E}_{\tau \sim p}[\nabla_\phi \log q] \tag{31}$$

$$\mathbb{E}_{\tau \sim p}[\nabla_\phi \mathcal{L}(\tau)] = -2\mathbb{E}_{\tau \sim p}[(\log p - \log q)\nabla_\phi \log q] \tag{32}$$

So why do we choose to minimize the joint matching loss rather than the (forward) KL divergence directly? This is because, as is clear from the above equations, it takes into account how far the 'predicted' log-probability $\log q$ currently is from $\log p$. That is, given a sample $\tau$, if $\log p < \log q$, the joint matching loss will actually *decrease* $\log q$. Instead, the forward KL will increase the probability for every sample it sees, and whether this particular sample will be too likely under $q$ can only be derived through sampling many trajectories.

Furthermore, we note that the joint matching loss is a 'pseudo' f-divergence with $f(t) = t \log^2 t$ [34]. It is not a true f-divergence since $t \log^2 t$ is not convex. A related well-known f-divergence is the Hellinger distance given by

$$H^2(p||q) = \frac{1}{2} \mathbb{E}_{\tau \sim p}[(\sqrt{p} - \sqrt{q})^2]. \tag{33}$$

This divergence similarly takes into account the distance between $p$ and $q$ in its derivatives through squaring. However, it is much less stable than the joint matching loss since both $p$ and $q$ are computed by taking the product over many small numbers. Computing the square root over this will be much less numerically stable than taking the logarithm of each individual probability and summing.

Finally, we note that we minimize the on-policy joint matching $\mathbb{E}_q[(\log p - \log q)^2]$ by taking derivatives $\mathbb{E}_q[\nabla_{\phi,\psi}(\log p - \log q)^2]$. This is technically not minimizing the joint matching, since it ignores the gradient coming from sampling from $q$.

## F  Dirichlet prior

This section describes how we fit the Dirichlet prior $p(\mathbf{P})$ used to train the inference model. During training, we keep a dataset of the last 2500 observations of $\mathbf{P} = f_\theta(\mathbf{x})$. We have to drop observations frequently because $\theta$ changes during training, meaning that the empirical distribution over $\mathbf{P}$s changes as well.

We perform an MLE fit on $k_W$ independent Dirichlet priors to get parameters $\boldsymbol{\alpha}$ for each. The log-likelihood of the Dirichlet distribution cannot be found in closed form [39]. However, since its log-likelihood is convex, we run ADAM [25] for 50 iterations with a learning rate of 0.01 to minimize the negative log-likelihood. We refer to [39] for details on computing the log-likelihood and alternative options. Since the Dirichlet distribution accepts positive parameters, we apply the softplus function on an unconstrained parameter during training. We initialize all parameters at 0.1.

We added L2 regularization on the parameters. This is needed because at the beginning of training, all observations $\mathbf{P} = f_\theta(\mathbf{x})$ represent uniform beliefs over digits, which will all be nearly equal. Therefore, fitting the Dirichlet on the data will give increasingly higher parameter values, as high parameter values represent low-entropy Dirichlet distributions that produce uniform beliefs. When the Dirichlet is low-entropy, the inference models learn to ignore the input belief $\mathbf{P}$, as it never changes. The L2 regularization encourages low parameter values, which correspond to high-entropy Dirichlet distributions.

## G  Designing symbolic pruners

We next discuss four high-level approaches for designing the optional symbolic pruner, each with differing tradeoffs in accuracy, engineering time and efficiency:

1. **Mathematically derive efficient solvers.** For simple problems, we could mathematically derive an exact solver. One example of an efficient symbolic pruner, along with a proof for exactness, is given for Multi-digit MNISTAdd in Appendix H. This pruner is linear-time. However, for most problems we expect the pruner to be much more computationally expensive.

2. **Use SAT-solvers.** Add the sampled symbols $\mathbf{y}$ and $\mathbf{w}_{1:i}$ to a CNF-formula, and ask an SAT-solver if there is an extension $\mathbf{w}_{i+1:k_W}$ that satisfies the CNF-formula. SAT-solvers are a general approach that will work with every function $c$, but using them comes at a cost.

   The first is that we would require grounding the logical representation of the problem. Furthermore, to do SAT-solving, we have to solve a linear amount of NP-hard problems. However, competitive SAT solvers can deal with substantial problems due to years of advances in their design [7], and a linear amount of NP-hard calls is a lower complexity class than #P hard. Using SAT-solvers will be particularly attractive in problem settings where safety and verifiability are critical.

3. **Prune with local constraints.** In many structured prediction tasks, we can use local constraints of the symbolic problem to prune paths that are guaranteed to lead to branches that can never create possible worlds. However, local constraints do not guarantee that each non-pruned path contains a possible world, but this does not bias the inference model, as the neural network will (eventually) learn when an expansion would lead to an unsatisfiable state.

   One example is the shortest path problem, where we filter out directions that would lead outside the $N \times N$ grid, or that would create cycles (See Appendix I.3). However, this only ensures we find *a* path, but does not ensure it is the shortest one.

4. **Learn the pruner.** Finally, we can learn the pruner, that is, we can train a neural network to learn satisfiability checking. One possible approach is to reuse the inference model trained on the belief $\mathbf{P}$ that uniformly distributes mass over all worlds.

   Learned pruners will be as quick as regular inference models, but are less accurate than symbolic pruners and will not guarantee that constraints are always satisfied during test-time. We leave experimenting with learning the pruner for future work.

## H   MNISTAdd Symbolic Pruner

In this section, we describe a symbolic pruner for the Multi-digit MNISTAdd problem, which we compute in time linear to $N$. Note that $\mathbf{w}_{1:N}$ represents the first number and $\mathbf{w}_{N+1:2N}$ the second. We define $n_1 = \sum_{i=1}^{N} w_i \cdot 10^{N-i-1}$ and $n_2 = \sum_{i=1}^{N} w_{N+i} \cdot 10^{N-i-1}$ for the integer representations of these numbers, and $y = \sum_{i=1}^{N+1} y_i \cdot 10^{N-i}$ for the sum label encoded by $\mathbf{y}$. We say that partial generation $\mathbf{w}_{1:k}$ has a *completion* if there is a $\mathbf{w}_{k+1:2N} \in \{0, \ldots, 9\}^{2N-k}$ such that $n_1 + n_2 = y$.

**Proposition H.1.** *For all $N \in \mathbb{N}$, $\mathbf{y} \in \{0,1\} \times \{0, \ldots, 9\}^N$ and partial generation $\mathbf{w}_{1:k-1} \in \{0, \ldots, 9\}^k$ with $k \in \{1, \ldots, 2N\}$, the following algorithm rejects all $w_k$ for which $\mathbf{w}_{1:k}$ has no completions, and accepts all $w_k$ for which there are:*

- *$k \leq N$: Let $l_k = \sum_{i=1}^{k+1} y_k \cdot 10^{k+1-i}$ and $p_k = \sum_{i=1}^{k} w_k \cdot 10^{k-i}$. Let $S = 1$ if $k = N$ or if the $(k+1)$th to $(N+1)$th digit of $y$ are all 9, and $S = 0$ otherwise. We compute two boolean conditions for all $w_k \in \{0, \ldots, 9\}$:*
$$0 \leq l_k - p_k \leq 10^k - S \tag{34}$$
*We reject all $w_k$ for which either condition does not hold.*

- *$k > N$: Let $n_2 = y - n_1$. We reject all $w_k \in \{0, \ldots, 9\}$ different from $w_k = \lfloor \frac{n_2}{10^{N-k-1}} \rfloor \bmod 10$, and reject all $w_k$ if $n_2 < 0$ or $n_2 \geq 10^N$.*

*Proof.* For $k > N$, we note that $n_2$ is fixed given $y$ and $n_1$ through linearity of summation, and we only consider $k \leq N$. We define $a_k = \sum_{i=k+2}^{N+1} y_i \cdot 10^{N+1-i}$ as the sum of the remaining digits of $y$. We note that $y = l_k \cdot 10^{N-k} + a_k$.

**Algorithm rejects $w_k$ without completions** We first show that our algorithm only rejects $w_k$ for which no completion exists. We start with the constraint $0 \leq l_k - p_k$, and show that whenever this constraint is violated (i.e., $p_k > l_k$), $\mathbf{w}_{1:k}$ has no completion. Consider the smallest possible completion of $\mathbf{w}_{k+1:N}$: setting each to 0. Then $n_1 = p_k \cdot 10^{N-k}$. First, note that
$$10^{N-k} > 10^{N-k} - 1 \geq a_k$$
Next, add $l_k \cdot 10^{N-k}$ to both sides
$$(l_k + 1) \cdot 10^{N-k} > l_k \cdot 10^{N-k} + a_k = y$$
By assumption, $p_k$ is an integer upper bound of $l_k$ and so $p_k \geq l_k + 1$. Therefore,
$$n_1 = p_k \cdot 10^{N-k} > y$$

Since $n_1$ is to be larger than $y$, $n_2$ has to be negative, which is impossible.

Next, we show the necessity of the second constraint. Assume the constraint is unnecessary, that is, $l_k > p_k + 10^k - S$. Consider the largest possible completion $\mathbf{w}_{k+1:N}$ by setting each to 9. Then
$$n_1 = p_k \cdot 10^{N-k} + 10^{N-k} - 1$$
$$= (p_k + 1) \cdot 10^{N-k} - 1$$
We take $n_2$ to be the maximum value, that is, $n_2 = 10^N - 1$. Therefore,
$$n_1 + n_2 = 10^N - (p_k + 1) \cdot 10^{N-k} - 2$$
We show that $n_1 + n_2 < y$. Since we again have an integer upper bound, we know $l_k \geq p_k + 10^k - S + 1$. Therefore,
$$y \geq (p_k + 1 + 10^k - S)10^{N-k} + a_k$$
$$\geq n_1 + n_2 + 2 - S \cdot 10^{N-k} + a_k$$

There are two cases.

- $S = 0$. Then $a_k < 10^{N-k} - 1$, and so

$$y \geq n_1 + n_2 + 2 + a_k > n_1 + n_2.$$

- $S = 1$. Then $a_k = 10^{N-k} - 1$, and so

$$y \geq n_1 + n_2 + 1 > n_1 + n_2.$$

**Algorithm accepts $w_k$ with completions** Next, we show that our algorithm only accepts $w_k$ with completions. Assume Equation 34 holds, that is, $0 \leq l_k - p_k \leq 10^k - S$. We first consider all possible completions of $\mathbf{w}_{1:k}$. Note that $p_k \cdot 10^{N-k} \leq n_1 \leq p_k \cdot 10^{N-k} + 10^{N-k} - 1$ and $0 \leq n_2 \leq 10^N - 1$, and so

$$p_k \cdot 10^{N-k} \leq n_1 + n_2 \leq (p_k + 1) \cdot 10^{N-k} + 10^N - 2.$$

Similarly,

$$l_k \cdot 10^{N-k} \leq y \leq (l_k + 1) \cdot 10^{N-k} - 1.$$

By assumption, $p_k \leq l_k$, so $p_k \cdot 10^{N-k} \leq l_k \cot 10^{N-k}$. For the upper bound, we again consider two cases. We use the second condition $l_k \leq 10^k + p_k - S$:

- $S = 0$. Then (since there are no trailing 9s),

$$
\begin{aligned}
y &\leq (l_k + 1) \cdot 10^{N-k} - 2 \\
&\leq (10^k + p_k + 1) \cdot 10^{N-k} - 1 \\
&= (p_k + 1) \cdot 10^{N-k} + 10^N - 2.
\end{aligned}
$$

- $S = 1$. Then with trailing 9s,

$$
\begin{aligned}
y &= (l_k + 1) \cdot 10^{N-k} - 1 \\
&\leq (10^k + p_k) \cdot 10^{N-k} - 1 \\
&= p_k \cdot 10^{N-k} + 10^N - 1 \\
&\leq (p_k + 1) \cdot 10^{N-k} + 10^N - 2,
\end{aligned}
$$

since $10^{N-k} \geq 1$.

Therefore,

$$p_k \cdot 10^{N-k} \leq y \leq (p_k + 1) \cdot 10^{N-k} + 10^N - 2$$

and so there is a valid completion.

$\square$

# I Details of the experiments

## I.1 Multi-digit MNISTAdd

Like [36, 37], we take the MNIST [30] dataset and use each digit exactly once to create data. We follow [36] and require more unique digits for increasing $N$. Therefore, the training dataset will be of size $60000/2N$ and the test dataset of size $10000/2N$.

### I.1.1 Hyperparameters

We performed hyperparameter tuning on a held-out validation set by splitting the training data into 50.000 and 10.000 digits, and forming the training and validation sets from these digits. We progressively increased $N$ from $N = 1$, $N = 3$, $N = 4$ to $N = 8$ during tuning to get improved insights into what hyperparameters are important. The most important parameter, next to learning rate, is the weight of the L2 regularization on the Dirichlet prior's parameters which should be very high. We used ADAM [25] throughout. We ran each experiment 10 times to estimate average accuracy, where each run computes 100 epochs over the training dataset. We used Nvidia RTX A4000s GPUs and 24-core AMD EPYC-2 (Rome) 7402P CPUs.

| Parameter name | Value | Parameter name | Value |
|---|---|---|---|
| Learning rate | 0.001 | Prior learning rate | 0.01 |
| Epochs | 100 | Amount beliefs prior | 2500 |
| Batch size | 16 | Prior initialization | 0.1 |
| # of samples | 600 | Prior iterations | 50 |
| Hidden layers | 3 | L2 on prior | 900.000 |
| Hidden width | 800 | | |

Table 4: Final hyperparameters for the multi-digit MNISTAdd task.

| Parameter name | Value | Parameter name | Value |
|---|---|---|---|
| Perception Learning rate | 0.00055 | Prior learning rate | 0.0029 |
| Inference learning rate | 0.003 | Amount beliefs prior | 2500 |
| Batch size | 20 | Prior initialization | 0.02 |
| # of samples | 500 | Prior iterations | 18 |
| Hidden layers | 2 | L2 on prior | 2.500.000 |
| Hidden width | 100 | | |
| Epochs | 5000 | Pretraining epochs | 50 |

Table 5: Final hyperparameters for the visual Sudoku puzzle classification task.

We give the final hyperparameters in Table 4. We use this same set of hyperparameters for all $N$. # of samples refers to the number of samples we used to train the inference model in Algorithm 1. For simplicity, it is also the beam size for the beam search at test time. The hidden layers and width refer to MLP that computes each factor of the inference model. There is no parameter sharing. The perception model is fixed in this task to ensure performance gains are due to neurosymbolic reasoning (see [35]).

### I.1.2 Other methods

We compare with multiple neurosymbolic frameworks that previously tackled the MNISTAdd task. Several of those are probabilistic neurosymbolic methods: DeepProbLog [35], DPLA* [37], NeurASP [58] and NeuPSL [44]. We also compare with the fuzzy logic-based method LTN [5] and with Embed2Sym [3] and DeepStochLog [56]. We take results from the corresponding papers, except for DeepProbLog and NeurASP, which are from [37], and LTN from [44][1]. We reran Embed2Sym, averaging over 10 runs since its paper did not report standard deviations. We do not compare DPLA* with pre-training because it tackles an easier problem where part of the digits is labeled.

Embed2Sym [3] uses three steps to solve Multi-digit MNISTAdd: First, it trains a neural network to embed each digit and to predict the sum from these embeddings. It then clusters the embeddings and uses symbolic reasoning to assign clusters to labels. A-NeSI has a similar neural network architecture, but we train the prediction network on an objective that does not require data. Furthermore, we train A-NeSI end-to-end, unlike Embed2Sym. For Embed2Sym, we use **symbolic prediction** to refer to Embed2Sym-NS, and **neural prediction** to refer to Embed2Sym-FN, which also uses a prediction network but is only trained on the training data given and does not use the prior to sample additional data

We believe the accuracy improvements compared to DeepProbLog to come from hyperparameter tuning and longer training times, as A-NeSI approximates DeepProbLog's semantics.

### I.2 Visual Sudoku Puzzle Classification

### I.2.1 A-NeSI definition

First, we will be more precise with the model we use. We see $\mathbf{x}$ as a $N \times N \times 784$ grid of MNIST images, and beliefs $\mathbf{P}$ as an $N \times N \times N$ grid of distributions over $N$ options (for example, for a $4 \times 4$ puzzle, we have to fill in the digits $\{0, 1, 2, 3\}$). For each grid index $i, j$, the world variable $\mathbf{w}_{i,j}$ corresponds to the digit at location $(i, j)$. For correct puzzles, we know that the digits at location

---

[1]We take the results of LTN from [44] because [5] averages over the 10 best outcomes of 15 runs and overestimates its average accuracy.

$(i, j)$ and location $(i', j')$ need to be different if $i = i'$, $j = j'$ or if $(i, j)$ is in the same block as $(i', j')$. For each pair of locations $(i, j), (i', j')$ for which this holds, we have a dimension in $\mathbf{y}$ that indicates if the digits at that grid location are indeed different. The symbolic function $c$ considers each such pair and returns the corresponding $\mathbf{y}$.

For the prediction model, we use a *single* MLP $f_{\boldsymbol{\theta}}$. That is, for each pair that should be different, we compute $q_{\boldsymbol{\phi}}(\mathbf{y}_k | \mathbf{P}) = f_{\boldsymbol{\phi}}(\mathbf{P}_{i,j}, \mathbf{P}_{i',j'})$. This introduces the independence assumption that the digits at location $(i, j)$ and location $(i', j')$ being different does not depend on the digits at other locations. This is, clearly, wrong. However, we found it is sufficient to accurately train the perception model.

When training the prediction model, since we sample $\mathbf{P}$ from a Dirichlet prior that assumes the different dimensions of $\mathbf{w}$ are independent, the grid of digits $\mathbf{w}$ are highly unlikely to represent actual sudoku's: There are about $10^2 1$ Sudoku's and $9^8 1$ possible grids (for $9 \times 9$ Sudoku's). However, it is quite likely to sample two digits that are different, and this is enough to train the prediction model.

### I.2.2  Hyperparameters and other methods

We used the Visual Sudoku Puzzle Classification dataset from [4]. This dataset offers many options: We used the simple generator strategy with 200 training puzzles (100 correct, 100 incorrect). We took a corrupt chance of 0.50, and used the dataset with 0 overlap (this means each MNIST digit can only be used once in the 200 puzzles). There are 11 splits of this dataset, independently generated. We did hyperparameter tuning on the 11th split of the $9 \times 9$ dataset. We used the other 10 splits to evaluate the results, averaging results over runs of each of those.

The final hyperparameters are reported in Table 5. The 5000 epochs took on average 20 minutes for the $4 \times 4$ puzzles, and 38 minutes for the $9 \times 9$ puzzles on a machine with a single NVIDIA RTX A4000. The first 50 epochs we only trained the prediction model to ensure it provides reasonably accurate gradients.

While [4] used NeuPSL, we had to rerun it to get accuracy results and results on $9 \times 9$ grids.

We implemented the exact inference methods using what can best be described as Semantic Loss [57]. We encoded the rules of Sudoku described at the beginning of this section as a CNF, and used PySDD (https://github.com/wannesm/PySDD) to compile this to an SDD [28]. This was almost instant for the $4 \times 4$ CNF, but we were not able to compile the $9 \times 9$ CNF within 4 hours, hence why we report a timeout for exact inference. To implement Semantic Loss, we modified a PyTorch implementation available at https://github.com/lucadiliello/semantic-loss-pytorch to compute in log-space for numerically stable behavior. This modified version is included in our own repository. We ran this method for 300 epochs with a learning rate of 0.001. We ran this method for fewer epochs because it is much slower than A-NeSI even on $4 \times 4$ puzzles (1 hour and 16 minutes for those 300 epochs, so about 63 times as slow).

For both A-NESI and exact inference, we train the perception model on *correct* puzzles by maximizing the probability that $p(y = 1 | \mathbf{P})$. A-NESI does this by maximizing $\log q_{\boldsymbol{\phi}}(\mathbf{y} = \mathbf{1} | \mathbf{P})$, while Semantic Loss uses PSDDs to exactly compute $\log p(y = 1 | \mathbf{P})$. For *incorrect* puzzles, there is not much to be gained since we cannot assume anything about $\mathbf{y}$. Still, for both methods we added the loss $-\log(1 - p(y = 1 | \mathbf{P}))$ for the incorrect puzzles.

### I.3  Warcraft Visual Path Planning

### I.3.1  A-NeSI definition

We see $\mathbf{x}$ as a $N \times N \times 3 \times 8 \times 8$ real tensor: The first two dimensions indicate the different grid cells, the third dimension indicates the RGB color channels, and the last two dimensions indicate the pixels in each grid cell. The world $\mathbf{w}$ is an $N \times N$ grid of integers, where each integer indexes five different costs for traversing that cell. The five costs are $[0.8, 1.2, 5.3, 7.7, 9.2]$, and correspond to the five possible costs in the Warcraft game. The symbolic function $c$ takes the grid of costs $\mathbf{w}$ and returns the shortest path from the top left corner $(1, 1)$ to the bottom right corner $(N, N)$ using Dijkstra's algorithm. We encode the shortest path as a sequence of actions to take in the grid, where each action is one of the eight (inter-)cardinal directions (down, down-right, right, etc.). The sequence is padded with the do-not-move action to allow for batching.

For the perception model, we use a single small CNN $f_{\boldsymbol{\theta}}$ for each of the $N \times N$ grid cells. That is, for each grid cell, we compute $\mathbf{P}_{i,j} = f_{\boldsymbol{\theta}}(\mathbf{x}_{i,j})$. The CNN has a single convolutional layer with 6 output dimensions, a $2 \times 2$ maxpooling layer, a hidden layer of $24 \times 84$ and a softmax output layer of $24 \times 5$, with ReLU activations.

The prediction model is a ResNet18 model [22], with an output layer of 8 options. It takes an image of size $6 \times N \times N$ as input. The first 5 channels are the probabilities $\mathbf{P}_{i,j}$, and the last channel indicates the current position in the grid. The 8 output actions correspond to the 8 (inter-)cardinal directions. We apply symbolic pruning (Section 3.3) to prevent actions that would lead outside the grid or return to a previously visited grid cell. We pretrained the prediction model by repeating Algorithm 1 on a fixed prior using 185.000 iterations (200 samples each) for $12 \times 12$, and 370.000 iterations (20 samples) for $30 \times 30$. We used fewer examples per iteration for the larger grid because Dijkstra's algorithm became a computational bottleneck. This took 23 hours for $12 \times 12$ and 44 hours for $30 \times 30$. Both used a learning rate of $2.5 \cdot 10^{-4}$ and an independent fixed Dirichlet prior with $\alpha = 0.005$. Standard deviations over 10 runs are reported over multiple perception model training runs on the same frozen pretrained prediction model. We trained the perception model for only 1 epoch using a learning rate of 0.0084 and a batch size of 70.

### I.3.2 Other methods

We compare to SPL [1] and I-MLE [41]. SPL is also a probabilistic neurosymbolic method, and uses exact inference. Its setup is quite different from ours, however. Instead of using Dijkstra's algorithm, it trains a ResNet18 to predict the shortest path end-to-end, and uses symbolic constraints to ensure the output of the ResNet18 is a valid path. Furthermore, it only considers the 4 cardinal directions instead of all 8 directions. SPL only reports a single training rule in their paper.

I-MLE is more similar to our setup and also uses Dijkstra's algorithm. It uses the first five layers of a ResNet18 to predict the cell costs given the input image. One big difference to our setup is that I-MLE uses continuous costs instead of a choice out of five discrete costs. This may be easier to optimize, as it gives the model more freedom to move costs around. I-MLE is reported using the numbers from the paper, and averages over 5 runs.

To be able to compare to another scalable baseline with the same setup, we added REINFORCE using the leave-one-out baseline (RLOO, [29]), implemented using the PyTorch library Storchastic [53]. It uses the same small CNN to predict a distribution over discrete cell costs, then takes 10 samples, and feeds those through Dijkstra's to get the shortest path. Here, we represent the shortest path as an $N \times N$ grid of zeros and ones. The reward function for RLOO is the Hamming loss between the predicted path and the ground truth path. We use a learning rate of $5 \cdot 10^{-4}$ and a batch size of 70. We train for 10 epoch and report the standard deviation over 10 runs. We note that RLOO gets quite expensive for $30 \times 30$ grids, as it needs 10 Dijkstra calls per training sample.

Finally, we experimented with running A-NeSI and RLOO simultaneously. We ran this for 10 epochs with a learning rate of $5 \cdot 10^{-4}$ and a batch size of 70. We report the standard deviation over 10 runs.

