# OpenReview forum: "A-NeSI: A Scalable Approximate Method for Probabilistic Neurosymbolic Inference"
_NeurIPS.cc/2023/Conference — NeurIPS 2023 poster_

### Official Review · Reviewer_8FBf · 2023-06-27

**Soundness:** 3 good
**Presentation:** 3 good
**Contribution:** 2 fair
**Rating:** 6
**Confidence:** 4

**Summary:**

This paper presents a variational framework for approximating neuro-symbolic inference. It addresses the weighted model counting (WMC) and most probable explanation (MPE) problems by introducing prediction and explanation models. Training techniques, such as output space factorization, regularized Dirichlet prior, and joint matching loss are discussed. To improve training efficiency and ensure logical constraint satisfaction during testing, a symbolic pruner is also proposed . Experimental evaluations demonstrate scalability and performance across three neuro-symbolic tasks.

**Strengths:**

- The paper is well-written and easy to understand, with a clear statement of the problem and methodology.
- The framework is simple and scalable, which may inspire further improvements in future work.
- Some training techniques are interesting, particularly the joint matching loss, which is related to GFlowNet.

**Weaknesses:**

- The authors have carefully discussed some weaknesses in Section 6.
- The motivation behind the prediction model and explanation model needs to be clarified.
    - It is unclear why the explanation model is necessary. From my understanding, only the prediction model is needed for neuro-symbolic training: we just need to train the prediction model using Eqn. (7), and then train the neural network using Eqn. (4); or repeat these two steps iteratively.
    - Another issue is the necessity of the prediction model. First, the prediction model is introduced to approximate weighted model counting, but approximate counting or sampling of $\mathbf{w}$ is still required during its training. Second, in Line 192, the authors report that the prediction model cannot perform better than using symbolic prediction.
- Some techniques are so specific that they appear to be designed only for the MNISTAdd task. For instance, as mentioned in Appendix G, the symbolic pruner is highly efficient for MNISTAdd task due to its simple mathematical structure. However, for more complex tasks, a SAT solver must be used, and the number of solver calls is proportional to the problem dimension, which is not acceptable.

**Questions:**

- Should  $\phi \leftarrow \textbf{Algorithm 2}(\textsf{beliefs}, \phi)$ be changed to $\textbf{Algorithm 1}$?
- Does the framework have requirements for background knowledge? For example, should it be presented in the form of CNF?
- Is it difficult to design and train the prediction model, given that it involves the approximation of the symbolic reasoning?

**Limitations:**

Yes, the limitations of the approach have been presented. The work does not have negative social impacts.

---

> ### Author Rebuttal · Authors · 2023-08-03
>
> We thank the reviewer for their comments and appreciate their support of the paper's writing and the framework's simplicity. The reviewer would like a clarification on the motivation behind the prediction and explanation model, which we provide below.
>
> > It is unclear why the explanation model is necessary. From my understanding, only the prediction model is needed for neuro-symbolic training: we just need to train the prediction model using Eqn. (7), and then train the neural network using Eqn. (4); or repeat these two steps iteratively.
>
> The reviewer is correct: The explanation model is indeed optional. The explanation model approximates the posterior, which can 'explain' a decision with the $\mathbf{w}$ that is most likely for that decision. As mentioned, interleaving Eq 7 and 4 (the prediction-only variant) is enough to train the perception model in A-NeSI: This is 'A-NeSI (predict)' in Table 1, and Tables 2 and 3 also use this variant.
>
> We will update the paper to emphasise this point.
>
> > Another issue is the necessity of the prediction model.
>
> The prediction model is necessary for A-NeSI and is why our method scales while exact inference does not. We will answer both arguments separately:
>
> > First, the prediction model is introduced to approximate weighted model counting, but approximate counting or sampling of $\\mathbf{w}$ is still required during its training.
>
> We do not perform approximate counting during the training of the prediction model. Instead, we indeed sample $\mathbf{w}$ to train it (see Equation 7). However, this sampling step is fast: To train the prediction model, we do not sample from the posterior $p(\mathbf{w}|\mathbf{y}, \mathbf{P})$ (expensive, requires counting in the normaliser) but from $p(\mathbf{w}|\mathbf{P})$ (fast, does not require normalising. It is just a single `torch.multinomial` call). Therefore, we bypass expensive counting or sampling while training the prediction model. Furthermore, evaluating $c(\mathbf{w})$ is also usually fast. We will highlight this efficiency in the paper.
>
> > Second, in Line 192, the authors report that the prediction model cannot perform better than using symbolic prediction.
>
> The reviewer is correct that during _testing_, the prediction model can not perform better than symbolic prediction in our tasks. However, the prediction model is still necessary: We need it to (efficiently/scalably) _train_ the perception model (Equation 4). However, we can throw the model away after training and use symbolic prediction for maximum accuracy. We will clarify this in the text.
>
> > Some techniques are so specific that they appear to be designed for the MNISTAdd task. For instance, the symbolic pruner is highly efficient for MNISTAdd task due to its simple mathematical structure. However, for more complex tasks, a SAT solver must be used, and the number of solver calls is proportional to the problem dimension, which is not acceptable.
>
> The symbolic pruner is very flexible in its design: While in the general case, we can always find a _perfect_ symbolic pruner with SAT-solving (which is, as you mention, quite expensive), we can often design fast imperfect pruners that still help A-NeSI. For instance, we used a fast, imperfect pruner in the Path Planning task to ensure the prediction model always returns a path (see Appendix I.3.1). We called those 'local constraints' in Appendix G.3. In hindsight, imperfect constraints may be more apt. Furthermore, as discussed in Appendix G.4, one could _learn_ an (imperfect) symbolic pruner.
>
> We will update Appendix G to clarify the possibilities of the design of the symbolic pruner.
>
> > Does the framework have requirements for background knowledge? For example, should it be presented in the form of CNF?
>
> Section 2.1 (problem components) defined the background knowledge as a deterministic black-box function $c$ between discrete input and output variables. Therefore, the two requirements are that $c$ is deterministic and that the domain and co-domain are discrete (although extensions to continuous variables are possible). The background knowledge does not have to be in a specific form (like CNF), but knowing about the form may help us design output factorisations and symbolic pruners. We will clarify this point in the background section.
>
> > Is it difficult to design and train the prediction model, given that it involves the approximation of the symbolic reasoning?
>
> This paper used simple MLPs for the first two experiments and a standard ResNet for the second. The training of the prediction models went smoothly for most experiments but the most challenging one (the 30x30 path planning). We trained on this problem for 44 hours and observed that the loss decreased slowly and still needed to converge.
>
> Does this generalise to more complex settings? There has been active research into whether neural networks can learn reasoning tasks, and the main consensus is positive as long as we evaluate the network in-distribution. We are not interested in out-of-distribution generalisation in A-NeSI, especially if the prior gives a good match. Furthermore, A-NeSI gives ample opportunity to design the prediction model with any neural network. In particular, GNNs will often be a good choice.
>
>
> > Should Algorithm 2 be changed to Algorithm 1?
>
> This is indeed a typing mistake. We will correct this in the revised version. Thanks!

---

> > ### Comment · Reviewer_8FBf · 2023-08-15
> > **Reply**
> >
> > Thank you for clarifying my concerns. However, I am still struggling to fully understand the motivation behind the prediction model. I know that $\mathbf{w}$ is sampled from $p(\mathbf{w} | \mathbf{P})$, and that evaluating $c(\mathbf{w})$ is often fast. However, finding a sample $\mathbf{w}$ that satisfies $c(\mathbf{w}) = \mathbf{y}$ can be difficult due to sparsity [1], and I think this is the main efficiency bottleneck. Could the authors elaborate more about how to address this sparsity issue?
> >
> > [1] Li, Q., Huang, S., Hong, Y., Chen, Y., Wu, Y. N., & Zhu, S. C. (2020, November). Closed loop neural-symbolic learning via integrating neural perception, grammar parsing, and symbolic reasoning.

---

> > > ### Author Response · Authors · 2023-08-15
> > >
> > > We thank the reviewer for elaborating on their concern. The reviewer is absolutely correct that finding a sample $\mathbf{w}$ for which $c(\mathbf{w})=\mathbf{y}$ is very difficult for a fixed $\mathbf{y}$. However,
> > > 1. Our algorithm (Algorithm 1 in the paper) does not search for a $\mathbf{w}$ that satisfies this constraint. Rather, we sample _any_ $\mathbf{w}$, evaluate $\mathbf{y}'=c(\mathbf{w})$ to see what output we get for this particular $\mathbf{w}$, and train the prediction network to learn this mapping. This loop is fast, and does not require any search.
> > > 2. However, the reviewer is right that  for the complex problems one will rarely sample a $\mathbf{w}$ that gives *exactly* the fixed $\mathbf{y}$ that we need. This is why we use the output factorization: indeed, the probability of sampling a $\mathbf{w}$ such that the output sum is exactly 362773882637293772 is extremely low. But sampling $\mathbf{w}$s for which some of the digits match is likely (around 1/10 for each digit). This allows us to train the prediction model efficiently.
> > > 3. Our experiments show that, with this loop, we can learn an approximate weighted model counter that is accurate enough to learn the perception model in complex problems. These problems could not be solved with other weighted model counting methods.
> > >
> > > We hope this addresses the concern and are available for further clarifications.

---

> > > > ### Comment · Reviewer_8FBf · 2023-08-15
> > > > **Reply**
> > > >
> > > > Thank you, I have no further remarks.

---

### Official Review · Reviewer_agwr · 2023-07-06

**Soundness:** 4 excellent
**Presentation:** 4 excellent
**Contribution:** 4 excellent
**Rating:** 7
**Confidence:** 3

**Summary:**

This paper introduces a variant of Probabilistic Neurosymbolic Learning that uses neural networks for approximate inference (vs. prior exponential-time exact inference). The efficacy of this approach is well-supported by various experiments.

**Strengths:**

- The paper is well-written and overall quite clear (i.e., the paper is high-quality in general). The goals are clearly stated. For example, Section 3 starts with "Our goal is to reduce the inference complexity of PNL."
- The experiments are described thoughtfully with good visualizations and tables.
- Using MNISTAdd as a running example throughout was a great idea. I think it added a lot.
- The authors are studying a significant problem -- i.e., that of neurosymbolic learning -- which feels especially timely.
- It is a bit harder for me to comment on originality. To the best of my knowledge, this work is rather original, but there might be other related prior works not mentioned by the authors of which I'm unaware.

**Weaknesses:**

My main concerns are about the notation. I think it could be improved, as discussed here and in the Questions section.
- See questions below.
- Beyond that, some of the notation is a bit clunky, particularly the double subscripts -- e.g., $q_{\phi_p}$.
- Relatedly, it seems like some of these subscripts are a bit inconsistent:
  - there's a p on the RHS of equation (2) but no p on the LHS of equation (3) - why?
  - in Section 3.3, there's both $\phi$ and $\phi_e$ seemingly inconsistently
  - more related questions below


**Questions:**

- The subscripts of $\textbf{P}$, the belief, seem a bit odd and not well-defined. Is $\textbf{P}$ a matrix? The subscript notation $_{w_i}$ seems especially odd/undefined. It would be good to make this more precise.
- Is it correct that, once you know $\textbf{P}$, you can just read off $p(\textbf{w}|\textbf{P})$ directly from $\textbf{P}$? If so, it seems like $p(\textbf{w}|\textbf{P})$ obscures this. Is there an alternative way of writing that to make it more explicit, like $f_P(w)$? (I don't love that notation, either - but I suspect there's a way to make this clearer. What do you think?)
- Is making the $\phi$ explicit in the notation accomplishing that much? My sense right now is that it's just obscuring the notation. Am I missing something?
- $k_W$ , $k_Y$ -- "discrete choices" maybe needs a bit more explanation. It's unclear at first whether this is the number of choices *within* $W$ -- i.e., $|W|$ -- or how many choices are drawn from $W$. Maybe specifying $k_W$ in the example -- e.g., $k_W = 2$ in the example -- would be helpful. Same for $k_Y$. Also, maybe this is the right place to introduce the notation $w_i$ for $i \in \{1,\dots,k_W\}$ and specify the spaces that $w_i$ and $y_i$ live in. If for some reason giving these space symbols is not needed or unnecessarily complicated, could you please explain? It seems more grounded to give them symbols.
- nit: should equation (5) have an indicator? (dropping it feels a bit too informal)
- The symbolic pruner notation feels off to me. Is there a separate $s$ for each $i$? Do $q,s$ depend on $i$ in $q \cdot s$? It seems like it should based on $s = s_i(\cdot)$, but it seems like it shouldn't if they're vectors (i.e., what are the dimensions of $s,q$?).
- Is $s_i$ given or learned?
- "In our studied tasks, neural prediction cannot perform better than symbolic prediction." But sudoku shows slightly better neural prediction than symbolic prediction. It would be good to comment on this.

**Limitations:**

The authors do a nice job outlining various limitations of this work. I found the presentation honest and straightforward.

---

> ### Author Rebuttal · Authors · 2023-08-03
>
> We thank the reviewer for their supportive comments. We appreciate the detailed feedback on notation and useful questions, which we discuss below.
>
> > Double subscripts on parameters are clunky, and their use is inconsistent
>
> You are correct. We will update this and make it more consistent throughout the text, likely by having different parameters $\mathbf{\phi}$ and $\mathbf{\psi}$ for the prediction and explanation models.
>
> We will also update the inconsistencies throughout.
>
> > The subscripts of the belief need to be more well-defined. Is it a matrix?
>
> Agreed, we will do this. $\mathbf{P}$ is not quite a matrix. It is a list where each element is in some simplex. If $\Delta_n$ is the n-dimensional simplex, then $\mathbf{P}_i\in \Delta\_{k\_{W\_i}}$, where $k\_{W\_i}$ is the number of categories in the $i$th variable of $W$. Therefore, each $\mathbf{P}_i$ represents a categorical probability distribution over $k\_{W\_i}$ options.
>
> It is like a matrix where the length of each row is variable, but this subtlety would make the notation dense, hence why we treat it informally as a matrix-like object. We will play around with this. Thanks!
>
> > The "discrete choices" need more explanation. Clarify their use in the example.
>
> Good question, and related to our answer from above. In an earlier version, we defined these formally, but we dropped this later to this for brevity, since the exact definition of the spaces is not used in the paper. We will reintroduce these definitions to balance these concerns.
>
> To answer the reviewer's question: Each $w_i\in \{1, ..., k_{W_i}\}$ is a categorical random variable. Here $k_{W_i}$ is the number of categories in the $i$th variable, like above. Then $k_W$ is the number of discrete variables in the problem.
>
> > Can you read of $p(\mathbf{w}|\mathbf{P})$ directly from $\mathbf{P}$? Is there a less obscuring notation?
>
> This is correct - And yes, it does obscure it somewhat. One option is to use $\mathsf{Cat}(\mathbf{w}; \mathbf{P})$ to emphasise it is a multivariate categorical distribution, but this is more verbose. We will play around with some options, thanks for the suggestion!
>
> > Is making the parameter $\phi$ explicit accomplishing much?
>
> We wanted to emphasise that $q$ is a neural network with some parameters that change during training, especially in the two algorithms. We will give this some thought, but will likely keep it.
>
> > nit: should equation (5) have an indicator? (dropping it feels a bit too informal)
>
> Correct, we (informally) dropped the indicator. We will reintroduce it.
>
> > The symbolic pruner notation feels off
>
> Looking back on this equation, we agree. We defined $\mathbf{s}$ as a $k\_{W\_i}$-dimensional vector, but (as you mention) it is much clearer to define $s$ as a function. Then the definition is $q\_{\phi\_e}(w\_i=j|\mathbf{y}, \mathbf{w}\_{1: i-1}, \mathbf{P})=\frac{\hat{q}\_{\phi\_{E}}(w_i=j|\mathbf{y}, \mathbf{w}_{1: i-1}, \mathbf{P}) s\_{\mathbf{y}, \mathbf{w}\_{1: i-1}}(w_i=j)}{\sum\_{j'=1}^{k\_{W\_i}} \hat{q}\_{\phi\_{E}}(w_i=j'|\mathbf{y}, \mathbf{w}\_{1: i-1}, \mathbf{P}) s\_{\mathbf{y}, \mathbf{w}\_{1: i-1}}(w\_i=j')}$ (where $s\_{\mathbf{y}, \mathbf{w}\_{1: i-1}}: W\_i \rightarrow \\{0, 1\\}$).
>
> > Is the symbolic pruner $s$ given or learned?
>
> In our experiments, we give $s$. In Appendix G, option 4 we describe an extension where $s$ is learned.
>
> > "In our studied tasks, neural prediction cannot perform better than symbolic prediction." But sudoku shows slightly better neural prediction than symbolic prediction. It would be good to comment on this.
>
> The variance of both of these results is rather high. We believe this discrepancy simply comes from a lack of statistical power, and both prediction methods perform about equally. We will mention this.

---

> > ### Comment · Reviewer_agwr · 2023-08-16
> >
> > Thanks for addressing my questions and committing to incorporating some of the proposed fixes! I maintain my score.

---

### Official Review · Reviewer_xznW · 2023-07-07

**Soundness:** 3 good
**Presentation:** 2 fair
**Contribution:** 2 fair
**Rating:** 7
**Confidence:** 4

**Summary:**

The paper presents a polynomial time solution to the approximate neurosymbolic inference problem for probabilistic neurosymbolic learning problems. The approach is based on a variant of predictive processing, with a prediction model and an explanation model. The results are interesting, and successfully solve three different non-trivial challenge problems: MNISTAdd, Visual Sudoku, and Warcraft Visual Path Planning.

**Strengths:**

+ The paper is an interesting attempt at bringing predictive processing concepts to neuro symbolic reasoning using probabilistic programs.
+ The idea of output space factorization is useful, as it helps reduce the complexity of the problem. However, it seems that humane effort may be needed to decompose the output in a helpful manner.
+ The design of belief priors using high-entropy Dirichlet distributions to cover all possible input scenarios or combinations of the factorized  space is an interesting observation.
+ The pseudocode in Figure 2 is clear and makes the underlying algorithm more easily reproducible.

**Weaknesses:**

- It is unclear how these experiments in controlled settings like MNISTAdd relate to robust neuro symbolic reasoning in more exciting tasks, like image classification or activity recognition. For example, factorization in such settings may not be easy.
- The paper has no discussion about how this approach is related to predictive processing. Perhaps, a comparison would be helpful to the readers of the paper.
- There are no bounds on the approximations being produced by this approach. How do the various design choices impact the quality of the approximation being obtained?

**Questions:**

- Does this approach lead to an improved algorithm or heuristic for weighted model counting in general?

Thanks for the response in the rebuttal.

**Limitations:**

The limitations listed in the paper are not clear. For example, it is noted that A-NESI did not perfectly train the prediction model in more challenging problems. It would be helpful to make this precise: what is the problem, what makes it challenging, and what was the quantitative performance of A-NESI? If it is one of the problems studied before, perhaps that section can be cited with an explanation.

The responses from the authors are interesting. I have updated my score to reflect my view and understanding of the paper.

---

> ### Author Rebuttal · Authors · 2023-08-03
>
> We thank the reviewer for their positive comments and interesting questions.
>
> > It is unclear how these experiments in controlled settings like MNISTAdd relate to robust neuro symbolic reasoning in more exciting tasks, like image classification or activity recognition. For example, factorization in such settings may not be easy.
>
> In the settings mentioned by the reviewer, we have background knowledge of what neural network outputs are allowed. For instance, consider scene graph generation. There, we may have rules like $\forall x, y: \phi(x, y)$. We have a natural decomposition over the universal quantifier: For each rule and every pair of objects, define a boolean variable with whether the proposition holds. This factorisation is rather large but should be easy to learn. We can also consider other heuristics, such as grouping per object and grouping by some rules.
>
> > The paper has no discussion about how this approach is related to predictive processing.
>
> We thank the reviewer for the suggestion. We are not experts in cognitive science and do not immediately see the suggested connection to predictive processing. However, the connection of the ideas of A-NeSI to cognitive science is an interesting option for future work. Of course, any suggested similarities from the reviewer are very welcome.
>
>
> > How do the various design choices impact the quality of the approximation being obtained?
>
> That is a good question. Our paper focused on empirically showcasing the strengths and scalability of our method, leaving approximation bounds for future work. However, we hypothesise these factors will make an impact: The expressiveness and architecture of the neural network, time of training, the divergence between the prior $p(\mathbf{P})$ and the evaluation distribution, and the complexity of the background knowledge. The latter include problem size, sparsity, connectedness, decomposability and structure. We highlighted some of these concerns in the limitations section and will give it another pass.
>
> Exactly how these relate is a complex question; we suspect no easy answer exists yet. There has been a significant amount of research into whether neural networks can learn reasoning tasks, in and out of distribution, and no clear and general answer exists.
>
> > Does this approach lead to an improved algorithm or heuristic for weighted model counting in general?
>
> Good question! Yes, we can use A-NeSI for general approximated WMC. However, we have yet to compare A-NeSI to existing approximation schemes: For this paper, our focus was on efficiently estimating the _gradient_ of the WMC, rather than on the accuracy of the WMC estimate itself.
>
> We hypothesise that the main benefit of A-NeSI comes from amortisation (like in variational inference): Training on many different values of $\mathbf{P}$ can be expensive but allows for rapid computation for a new value of $\mathbf{P}$. Furthermore, A-NeSI is very flexible in computation budgets: We can train for as many iterations as possible in our compute budget to improve our estimations. A downside of A-NeSI is the lack of guarantees on the approximation bounds, as the reviewer mentions. Future work could study this question.
>
> > The limitations listed are unclear. Make this precise: what is the problem, what makes it challenging, and what was the quantitative performance of A-NESI?
>
> The fact that the prediction model is not trained perfectly is made clear in the divergence between neural and symbolic predictions in Section 4.1 for $N=15$. If there is such a divergence, the most likely answer, according to the prediction model, is not the one reflected by the MAP sum. Here, the increased dimensionality made it harder for the neural network to learn the problem perfectly. We found similar issues in path planning, a much more complicated task in all three dimensions mentioned in our limitations section: We needed much longer training times to get good gradient estimates. We revised the paper to add these examples.

---

### Official Review · Reviewer_iEEt · 2023-07-08

**Soundness:** 4 excellent
**Presentation:** 4 excellent
**Contribution:** 3 good
**Rating:** 7
**Confidence:** 4

**Summary:**

This paper introduces A-NeSi, a fast approximate procedure for NeSy architectures
based on probabilistic logic.  These architectures do not scale gracefully to
problems involving many possible worlds, and the goal of A-NeSi is to scale them
up.  In short, A-NeSi employs a surrogate models (autoregressive neural nets)
to replace the most computationally intensive steps.  Experiments indicate a
sizeable speed-up compared to SOTA probabilistic-logical approaches on a
variety of tasks - from multidigit MNIST addition to pathfinding.

**Strengths:**

+ Very clearly written.  There's plenty of figures to build intuition, which is good.
+ Tackles a very prominent problem in neuro-symbolic AI - scalability of MAP inference for probabilistic-logical approaches, a major player in this field.
+ The idea of using surrogates is simple but very sensible.  (What is not simple is ensuring the output satisfies the knowledge.)
+ All modeling choices are clearly motivated.
+ Empirical evaluation is both extensive and very promising.
+ Related work is both compact and comprehensive, well done.

**Weaknesses:**

No major deficiencies that I could find.  This is a solid contribution.

**Questions:**

Q1. How strongly does performance of A-NeSi depend on the complexity of the hard constraints/knowledge?  Especially when knowledge encodes long-range interactions between variables (or, if you prefer, tree width).  I would imagine beam search could choke up in these cases, and I'd appreciate some discussion on the expected behavior.

**Limitations:**

Limitations are clearly discussed in the conclusion.

---

> ### Author Rebuttal · Authors · 2023-08-03
>
> We thank the reviewer for their positive comments.
>
> > How strongly does performance of A-NeSi depend on the complexity of the hard constraints/knowledge? Especially when knowledge encodes long-range interactions between variables (or, if you prefer, tree width). I would imagine beam search could choke up in these cases, and I'd appreciate some discussion on the expected behavior.
>
> Good question. We believe the complexity of the constraints matters significantly for the learnability of the prediction model. While we leave a proper study for future work, we predict the task's predictability, decomposability and degree of structure are important factors. Addition and sudokus are two tasks that are easy to decompose, making learning the prediction model relatively easy. Path planning is less decomposable and, as you mention, requires more long-range interactions. The prediction model took much longer to train for this task.
>
> The reviewer proposes the performance of beam search may degrade in challenging tasks. We do not currently see the bottleneck in the (test-time) beam search: In most neurosymbolic tasks, the perception model has low entropy after training, meaning there are few options to consider. For instance, [1] found that finding the MAP state is efficient in low-entropy distributions.
>
> [1] Manhaeve, R., Marra, G., & De Raedt, L. (2021). Approximate inference for neural probabilistic logic programming. In Proceedings of the 18th International Conference on Principles of Knowledge Representation and Reasoning (pp. 475-486). IJCAI Organization.

---

> > ### Comment · Reviewer_iEEt · 2023-08-12
> > **Reply**
> >
> > Thanks for answering my question.  I would appreciate if the authors could mention their observations re. long-range interactions and low entropy/beam search performance in the paper.  I remain of the opinion that this work is well worth accepting.

---

> > > ### Author Response · Authors · 2023-08-14
> > >
> > > Thanks for the comment. We agree, and will add it to the camera ready. We thank the reviewer for the suggestion.

---

### Author Rebuttal · Authors · 2023-08-03

We thank the reviewers for their time and their reviews. The reviewers mention the clear writing and statement of the problem, motivation and methodology (`iEET`, `agwr`, `8FBf`), in particular, the use of the MNISTAdd example (`agwr`), the comprehensiveness of related work (`iEEt`), the description of the experiments (`agwr`) and the pseudocode (`xznW`). The reviewers also mention the importance of the problem A-NeSI tackles (`iEEt`, `agwr`). The reviewers discuss the simplicity of the method (`iEEt`, `8FBf`) with possibilities for future work (`8FBf`) while mentioning that A-NeSI contains several interesting (`xznW`, `8FBf`) and original (`agwr`) ideas. Furthermore, the reviewers note that the empirical evaluation is extensive and promising (`iEEt`), demonstrating scalability and performance (`agwr`, `8FBf`) in non-trivial problems (`xznW`).

Reviewer `8FBf` asked for clarification on the motivation of the prediction and explanation models. We agree with reviewer `8FBf` that the explanation model is optional and discuss the benefits of also modelling the explanations. Furthermore, we clarify why training the prediction model does not require approximate counting or expensive sampling. Finally, while at _test-time_ the prediction model cannot be better than the symbolic prediction, we discuss why we need the prediction model to do scalable _training_.

Reviewer `agwr` provides valuable in-depth comments about the notation, most of which we implement. In different forms, the reviewers `xznW`, `iEEt` and `8FBf` all ask how the techniques proposed in A-NeSI will generalise to other settings. We answer or clarify these questions in a separate response to the individual reviewers.

---

### Decision · Program_Chairs · 2023-09-21

**Decision:**

Accept (poster)

**Comment:**

This paper aims to scale up neurosymbolic methods by learning predictive models that can stand in for a symbolically computed loss function. The paper addresses an important problem in neurosymbolic AI and the proposed solution performs reasonably well.